# Learning a Zeroth-Order Optimizer for Fine-Tuning LLMs

**Kairun Zhang** [* 1]   **Haoyu Li** [* 1]   **Yanjun Zhao** [* 1]   **Yifan Sun** [1]   **Huan Zhang** [1]

## Abstract

Zeroth-order optimizers have recently emerged as an attractive approach for fine-tuning large language models (LLMs), as they avoid backpropagation and can substantially reduce memory overhead relative to standard first-order training. However, existing zeroth-order methods rely on hand-crafted, static sampling strategies that are not adaptable to model-specific structures. To address this, we propose ZO Fine-tuner, a learning-based zeroth-order optimizer for LLMs that automatically learns efficient perturbation strategies through a compact and memory-efficient design. Motivated by the fact that a small set of base LLMs is repeatedly fine-tuned across tasks, ZO Fine-tuner supports one-time per-model training and reuse across downstream tasks with minimal overhead. Therefore, learning the optimizer once for a given LLM and reusing it across diverse downstream tasks is both feasible and highly desirable. Accordingly, ZO Fine-tuner is designed to scale learning to learn (L2L) to the foundation-model era by supporting one-time per-model training with minimal overhead. Experiments on 4 LLMs and 7 datasets show that ZO Fine-tuner outperforms prior zeroth-order baselines in 82.1% of task-model combinations, thereby demonstrating strong performance and scalability for efficient LLM fine-tuning. The code can be found in `https://github.com/ASTRAL-Group/ZO_Fine_tuner`.

## 1. Introduction

Nowadays, fine-tuning pre-trained foundation models on downstream tasks has become a standard paradigm. However, as model sizes grow, traditional first-order optimizers such as Adam become increasingly expensive. In particular,

these methods impose significant memory overhead, up to 12 times (Malladi et al., 2023) more than inference. Even with parameter-efficient fine-tuning (PEFT) methods such as LoRA (Hu et al., 2022) and Prefix-Tuning (Li & Liang, 2021), training can still incur nontrivial memory needs due to backpropagation and optimizer states.

To address these challenges, memory-efficient zeroth-order (MeZO) optimizer (Malladi et al., 2023) has been proposed. This approach only requires two forward passes per step and achieves competitive performance to first-order methods while maintaining memory usage comparable to inference. Many subsequent methods, such as HIZOO (Zhao et al., 2025), LOZO (Chen et al., 2024), MeZO-SVRG (Gautam et al., 2024), ZO-AdamU (Jiang et al., 2023), and ZO-DAP (Ma & Huang, 2025) attempt to improve upon MeZO by manually designing more sophisticated parameter-updating rules. However, these designs are often based on intuition or mathematical approximations, and still typically require extensive hyperparameter tuning beyond learning rates to perform well in practice.

We argue that prior works have largely overlooked the potential of learning to learn (L2L) techniques (Andrychowicz et al., 2016) for improving zeroth-order optimization in this setting. Unlike hand-designed optimizers, L2L provides a data-driven approach to learn effective optimization strategies. Rather than manually tuning update rules and hyperparameters, L2L leverages auxiliary neural networks that adaptively guide the optimization process. These learned optimizers typically rely on the same information accessible to conventional optimizers, such as gradient signals or their approximations. By leveraging such inputs, they often outperform manually designed counterparts in both convergence speed and final performance, as they are able to explore the loss landscape more effectively (Wichrowska et al., 2017a). For example, learned optimizers have been shown to surpass SGD and even Adam across a variety of models and tasks (Lv et al., 2017a). Similar improvements have also been observed in zeroth-order optimization settings on small-scale models (Ruan et al., 2020).

While L2L methods have shown promise on small-scale models (Chen et al., 2021), we believe their potential is even greater in the era of foundation models. In the small-model regime, different tasks typically require different

---
[*]Equal contribution [1]University of Illinois Urbana-Champaign. Correspondence to: Haoyu Li <haoyuli5@illinois.edu>, Huan Zhang <huan@huan-zhang.com>.

*Proceedings of the 43$^{rd}$ International Conference on Machine Learning*, Seoul, South Korea. PMLR 306, 2026. Copyright 2026 by the author(s).

models, and L2L optimizers often exhibit limited transferability across model architectures (Wichrowska et al., 2017a). As a result, a separate optimizer must be trained for each model-task pair, leading to substantial additional costs. In contrast, a recent LLM supply chain study shows that while there are many specialized checkpoints on platforms like Huggingface, most are derivatives of a handful of core base models like Llama and Qwen (Shahedur Rahman et al., 2025). Moreover, for a given LLM, the structure or properties leveraged by certain optimizers are often consistent across tasks (Guo et al., 2024). This provides a great opportunity for L2L methods, where **a learned optimizer trained once for a base LLM can be potentially reused across diverse derivative models and tasks**. Toward practical adoption, if model creators were to ship a pretrained learned finetuner alongside each base model, it would unlock a memory-efficient fine-tuning path with competitive performance for downstream users.

In the context of zeroth-order optimization for LLMs, learning a perturbation distribution with non-uniform and adaptive variance scales, instead of a standard normal distribution, could be beneficial (Ye et al., 2018; Gao & Sener, 2022; Zhao et al., 2025). However, the sheer number of parameters of LLMs introduces new challenges when applying L2L as it requires differentiating through the optimization process, which demands storing a substantial number of activations for backpropagation. Moreover, naively applying coordinate-wise auxiliary networks at the LLM scale can result in prohibitive memory overhead. To address this, we draw inspiration from recent analyses of Transformer curvature (Zhang et al., 2024b), which suggest that Transformer Hessians often exhibit an approximately block-diagonal structure. This inspires us to adopt a block-wise parameterization and share a single perturbation variance within each parameter block. We thus propose ZO Fine-tuner, which consists of highly compact and memory-efficient per-parameter-block auxiliary networks that learn shared effective perturbation variances. As a result, the **memory cost is minimal**: for OPT-30B, the storage required for all auxiliary networks is less than 2MB under FP16 precision, which is negligible compared to the 60GB model itself. In the meantime, through extensive experiments, we demonstrate that our ZO Fine-tuner **trained on a single** dataset is **highly generalizable across model derivatives and datasets**, which strongly underscores the potential of the "train once, reuse widely" goal. Our contributions are summarized as follows:

• We extend the L2L framework to LLMs and show that a single learned optimizer trained on a base model can generalize across downstream tasks and derivative checkpoints.

• We propose block-wise perturbations with a shared variance per parameter block, enabling L2L at LLM scale.

• Across four models and seven datasets (28 task-model pairs), ZO Fine-tuner outperforms the strongest baseline (lower training loss) in 82.1% of the combinations, achieving an average of 2.5% improvement in accuracy with tiny memory and time overhead.

## 2. Related Work

**Zeroth-order optimization.** Zeroth-order optimization appears in a wide range of applications where either the objective function is implicit, or its gradient is impossible or too expensive to compute. For example, methods such as (Tang et al., 2021; Hajinezhad & Zavlanos, 2018) consider derivative-free distributed algorithms for non-convex multi-agent optimization. ZO-BCD (Cai et al., 2021), ZOO (Chen et al., 2017), ZO-signSGD (Liu et al., 2019), and ZO-HessAware (Ye et al., 2019) utilize zeroth-order stochastic optimization to generate black-box adversarial examples in deep learning. Beyond that, MeZO (Malladi et al., 2023) firstly adapted the classical ZO-SGD method to fine-tune LLMs, while achieving comparable performance with extremely great memory reduction. Subsequently, ZO-AdaMU (Jiang et al., 2023) improved ZO-SGD by incorporating momentum into its stochastic approximation process. HIZOO (Zhao et al., 2025) leverages Hessian information to enhance performance in a memory-efficient manner. Other works explore structural properties of the gradient to improve MeZO, such as utilizing low-rank approximations (Chen et al., 2024; Sun et al., 2025) or exploiting gradient sparsity (Guo et al., 2024; Liu et al., 2024).

**Learning to learn.** Previous studies have investigated using neural networks to improve optimization update rules, replacing manually crafted algorithms such as Adam (Kingma & Ba, 2015). (Cotter & Conwell, 1990) tried to use recurrent neural networks (RNNs) to model the optimization process to learn adaptively. After that, (Baxter, 1998) gave an overview of the idea and techniques of learning to learn; for example, they proposed to train RNNs to optimize basic convex functions. Then (Andrychowicz et al., 2016; Wichrowska et al., 2017b; Metz et al., 2019; 2022; Lv et al., 2017b) introduced a variety of sophisticated strategies to enhance the performance of optimizers in deep learning. Additionally, (Li & Malik, 2016) and (Li & Malik, 2017) adopted reinforcement learning (RL) policy search techniques into the L2L framework. In the context of zeroth-order optimization, (Ruan et al., 2020) applied L2L techniques to enhance performance on small-scale models.

## 3. Method

### 3.1. Motivation

The foundational work in zeroth-order optimization for LLM fine-tuning, MeZO (Malladi et al., 2023), simply esti-

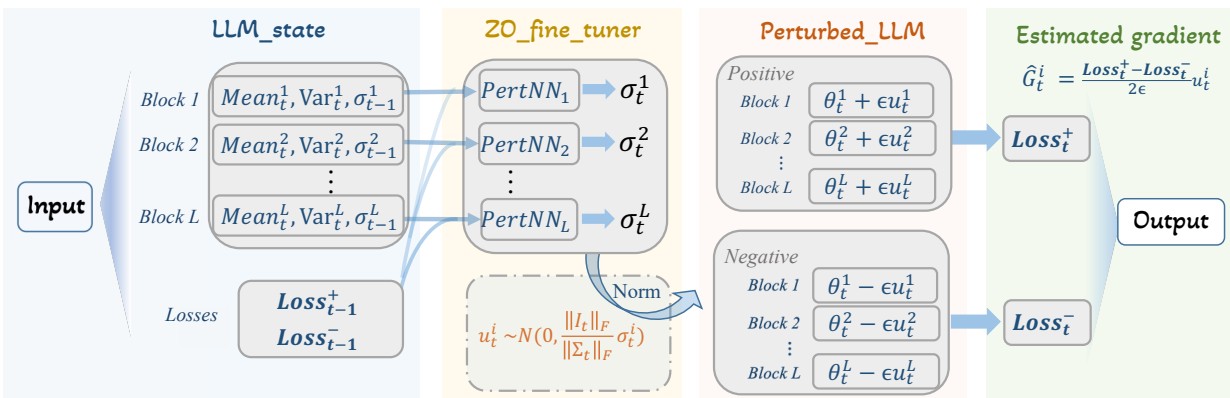

*Figure 1.* Fine-tune the LLM using trained ZO Fine-tuner. Each block of the LLM is equipped with a lightweight neural network that predicts its perturbation variance. For LLM parameter $\theta_t^i$ in block $i$ at step $t$, $\text{PertNN}_i$ takes in compact summarizing statistics the $\text{Mean}_t^i$, $\text{Var}_t^i$ of the $\theta_t^i$. Additionally, it takes in the last perturbation variance $\sigma_{t-1}^i$, and the two losses recorded at the last step. It outputs the updated perturbation variance $\sigma_t^i$ and then applies normalization. By learning non-uniform, layer-specific perturbation scales and plugging them into standard zeroth-order updates, the fine-tuner enables efficient, high-performance gradient-free optimization of LLM.

mates the directional derivative as the step size on a certain sampled direction by evaluating the model at two perturbed parameter points. This approach only requires two forward passes and avoids backpropagation, making it attractive for memory-constrained training. Given a model with parameters $\theta \in \mathbb{R}^d$ and loss function $\mathcal{L}$, MeZO estimates the gradient on a mini-batch $\mathcal{B}_t$ as:

$$\hat{g}(\theta_t; \mathcal{B}_t) = \frac{\mathcal{L}(\theta_t + \epsilon u_t; \mathcal{B}_t) - \mathcal{L}(\theta_t - \epsilon u_t; \mathcal{B}_t)}{2\epsilon} u_t, \quad (1)$$

$$u_t \sim \mathcal{N}(0, I_d),$$

We argue that a fixed sampling distribution such as $\mathcal{N}(0, I)$ is suboptimal: the quality of zeroth-order gradient estimates depends on local properties of the landscape at each step (Ye et al., 2018; Gao & Sener, 2022; Zhao et al., 2025). Therefore, through learning, L2L approach has the potential to generate perturbations $u_t$ that are informed by such local signals and thus allocate perturbation effort more effectively. However, naive implementations of this idea can incur prohibitive memory overhead. For example, learning a separate perturbation for each individual parameter using a fully connected auxiliary network would require at least $O(d^2)$ parameters for a model with $d$ parameters.

We thus turn to exploit geometric structure in LLMs. Recent empirical analyses suggest that Transformer curvature often exhibits an approximately block-structured pattern, where much of the Hessian mass concentrates within natural parameter groups (e.g., embeddings, attention Q/K/V matrices, and projections) (Zhang et al., 2024b). This observation motivates a coarse-grained control of perturbations: instead of learning coordinate-wise perturbations, we target per-block adaptation. We now formalize this idea and show that even a simple, block-wise adjustment of perturbation variance can improve over MeZO.

**Theorem 3.1** (Informal Version). *Define the expected change in loss after performing a one step update in parameter $\theta_t$ as $d(\theta_t) := \mathbb{E}\left[\mathcal{L}(\theta_{t+1}) \mid \theta_t\right] - \mathcal{L}(\theta_t)$. Suppose now the Hessian matrix $H(\theta_t)$ is block-diagonal $H(\theta_t) := \text{diag}(H_1(\theta_t), \cdots, H_b(\theta_t))$, then by varying the $\sigma_i$'s in $\Sigma := \text{diag}(\sigma_1 I, \cdots, \sigma_b I)$, the same gradient estimation (1) but with $u_t \sim \mathcal{N}(0, \Sigma\Sigma^\top)$ can yield tighter upper bound on $d(\theta_t)$ compared to MeZO.*

The formal version and proof of this theorem can be found in appendix D. At a high level, this theorem states that if the Hessian exhibits a block wise structure, then learning an adaptive per-block shared variance can improve convergence over MeZO. Crucially, the per-block parameterization yields this improvement *without* incurring prohibitive memory cost as the number of parameter blocks is far less than the number of parameters. For instance, in LLaMA-8B, the model contains only 291 parameter blocks, despite having over 8 billion individual parameters. This result thus motivates and justifies our design of ZO Fine-tuner, a per-block variance learner. Below, we first discuss the architecture of our ZO Fine-tuner and how to finetune downstream LLMs using a given ZO Fine-tuner. Then we introduce the training scheme for ZO Fine-tuner to enable generalizations.

### 3.2. ZO Fine-tuner

**Architecture.** As we discussed in the motivation section, we design ZO Fine-tuner to dynamically generate a block diagonal variance matrix $\Sigma_t$ corresponding to each parameter group at each optimization step via lightweight neural networks named PertNN. To incorporate all the dynamic information, PertNN takes in model parameters $\theta_t$, previously used perturbation variances $\Sigma_{t-1}$, and their observed loss as inputs $\ell_{t-1}$. Intuitively, these inputs encourage PertNN to consider the effectiveness of past updates, where perturba-

tions that lead to sharper loss changes might indicate more informative directions. However, we notice that the model parameters are still too memory-intensive as an input feature. Therefore, we further compress the memory usage by only feeding the summarizing statistics of the model parameters $\theta_t$ into PertNN, such as $\text{Mean}(\theta_t)$ and $\text{Var}(\theta_t)$.

Formally, the perturbation variance at each step $t$ is generated as follows. For parameter block $i$, $\sigma_{t-1}^{(i)}$ is the previous perturbation variance, $d_i$ is the number of parameters in this block, and $\text{Mean}_t^{(i)}$ and $\text{Var}_t^{(i)}$ represent the current mean and variance of the block's parameter values. $\omega^{(i)}$ denotes the learnable parameters of the auxiliary neural network assigned to block $i$.

$$
\sigma_t^{(i)} = \text{PertNN}^{(i)}\left(\boldsymbol{\ell}_{t-1}, \sigma_{t-1}^{(i)}, \text{Mean}_t^{(i)}, \text{Var}_t^{(i)}; \omega^{(i)}\right),
$$
$$
\Sigma_t = \text{diag}(\sigma_t^{(1)}I_{d_1}, \sigma_t^{(2)}I_{d_2}, \ldots, \sigma_t^{(n)}I_{d_n}).
$$
(2)

With this variance, ZO Fine-tuner then updates model parameters with

$$
\hat{g}(\theta_t; \mathcal{B}_t; \omega) = \frac{\mathcal{L}(\theta_t + \epsilon u_t; \mathcal{B}_t) - \mathcal{L}(\theta_t - \epsilon u_t; \mathcal{B}_t)}{2\epsilon} u_t,
$$
$$
u_t \sim \mathcal{N}(0, \Sigma_t\Sigma_t^\top), \quad \theta_{t+1} = \theta_t - \eta\,\hat{g}(\theta_t; \mathcal{B}_t; \omega).
$$
(3)

Importantly, we should note that $\hat{g}$ is inherently a function of $u_t$, which is a function of $\Sigma_t$, and thus a function of the parameter of PertNN $\omega$. To enable gradient-based training of PertNN within the L2L framework, we adopt the reparameterization trick: instead of sampling $u_t$ directly from $\mathcal{N}(0, \Sigma_t)$, we sample $z_t \sim \mathcal{N}(0, I_d)$ and compute $u_t = \Sigma_t z_t$. This makes the entire perturbation process differentiable, allowing gradients to flow back through the perturbation generation module.

**Normalization.** Although effective, this non-uniform variance introduced a new challenge when using ZO Fine-tuner as an optimizer. From the two-point ZO estimator, we see

$$
\mathbb{E}[\hat{g}(\theta_t; \mathcal{B}_t)] = \mathbb{E}\Big[\frac{\mathcal{L}(\theta_t + \varepsilon u_t; \mathcal{B}_t) - \mathcal{L}(\theta_t - \varepsilon u_t; \mathcal{B}_t)}{2\varepsilon}\Big]
$$
$$
\approx \mathbb{E}[u_t u_t^\top]\nabla\mathcal{L}(\theta_t; \mathcal{B}_t).
$$
(4)

Therefore, when fine-tuning downstream tasks, we note that the effective learning rate became $\eta \cdot \frac{\|u_t\|^2}{d}$ on average. This makes controlling the effective learning rate difficult, and the learned variance $\Sigma_t$ became a confounding variable in the update size. In reality, we wish $\Sigma_t$ to only carry information about relative block-wise variance, and we could still use a single learning rate to control the overall step size to ensure stable training. Therefore, we introduce the following normalization, which ensures the decoupling of the variance and the learning rate. We note that if $u_t = \Sigma_t z_t$ with $z_t \sim \mathcal{N}(0, I)$, then $\mathbb{E}\|u_t\|^2 = \text{tr}(\Sigma_t\Sigma_t^\top) = \|\Sigma_t\|_F^2$. We then normalize by fixing the total variance budget and let

$\|\Sigma_t\|_F^2 = \|I_d\|_F^2 = d$. Thus, only the relative block-wise variances are learned. In practice, this keeps $\|u_t\|$ approximately constant (by concentration in high dimensions). For example, with our generated $\Sigma_t$, if $u_t \sim \mathcal{N}(0, \Sigma_t\Sigma_t^\top)$, $\|u_t\|$ concentrates around $\|\Sigma_t\|_F$ and we achieve the desired control over the effective learning rate.

**Complete Optimization Algorithm.** As summarized in Algorithm 1 and Figure 1, ZO Fine-tuner first compute the block-wise non-uniform perturbation variance $\Sigma_t$ using the learned neural network PertNN. Then it applies normalization to control the overall magnitude of the perturbation. Finally, it uses the normalized perturbation to update the LLM following (3). We notice this incurs minimal overhead compared to MeZO, in terms of both memory and speed. In particular, the only memory overhead compared to MeZO is the light-weight per-block PertNN, whereas the only speed overhead is the query to PertNN.

---

**Algorithm 1** Finetuning a LLM with ZO Fine-tuner

**Input:** LLM parameters $\theta$, PertNN parameters $\omega$, training step $T$, learning rate $\eta$.
Initialize variance $\Sigma_0$ as $I_d$, LLM parameter as $\theta_0$.
**for** $t = 1, ..., T$ **do**
  Sample a batch $\mathcal{B}_t$ from $\mathcal{T}$
  $\Sigma_t \leftarrow \text{PertNN}(\theta_t, \Sigma_{t-1}, \boldsymbol{\ell}_{t-1}; \omega)$
  Sample $z_t \sim \mathcal{N}(0, I_d)$, $\widetilde{\Sigma}_t \leftarrow \frac{\|I_d\|_F}{\|\Sigma_t\|_F}\Sigma_t$
  $u_t \leftarrow \widetilde{\Sigma}_t z_t$
  Compute loss with perturbed parameter to obtain $\ell_t$
  $\hat{g}_t = \frac{l_t^+ - l_t^-}{2\epsilon}u_t$
  $\theta_{t+1}^t = \theta_t - \eta\hat{g}_t$
**end for**

---

### 3.3. Training ZO Fine-tuner

We now turn to training ZO Fine-tuner in a L2L fashion. The key idea is to treat the model's own finetuning trajectory as supervision. After a single update by ZO Fine-tuner, we evaluate the post-update loss and adjust PertNN so as to reduce this quantity across tasks. We next formalize this meta-objective and outline several practical choices that make training stable.

**Data Source and Objective Function.** First, we need a source of training data for our ZO Fine-tuner. In our setting, this data corresponds to different model states with various losses. A key insight of us is to notice that the fine-tuning process of LLMs under a first-order optimizer naturally produces a trajectory of intermediate model states, and we can directly leverage this trajectory to optimize the perturbation variance generator.

Along the first order optimization trajectory with loss function $\mathcal{L}$, we obtain a set of model parameters $\{\theta_0^k\}_k$. We then attempt to perform a one-step zeroth-order update using

our ZO Fine-tuner with update rule 3 to get $\theta_1^k$ and use the resulting loss as a feedback signal to assess and optimize the effectiveness of the current perturbation strategy. Specifically, at each step we aim to minimize the post-update loss $\mathcal{L}(\theta_1^k)$. As we discussed in section 3.2, the estimated gradient $\hat{g}$ is implicitly a differentiable function of the parameters $\omega$ of PertNN per the reparametrization trick. Therefore, we can use a gradient-based method to update ZO Fine-tuner. Formally, the objective for training ZO Fine-tuner is therefore:

$$\min_\omega \mathcal{L}_{\text{ZO}}(\theta_0^k; \omega) := \min_\omega \mathcal{L}\left(\theta_0^k - \eta \hat{g}(\theta_0^k, \omega)\right) \quad (5)$$

After the update, we move the parameters $\theta_0^k$ along the first-order trajectory to get $\theta_0^{k+1}$ and continue learning. As the inputs to ZO Fine-tuner are task and model-agnostic state summaries, rather than task-specific features, the learned decisions are largely invariant to differences across datasets or nearby checkpoints. As we will demonstrate in experiments, our ZO Fine-tuner trained on one single dataset can be transferred to efficiently finetune other datasets and model derivatives.

**Periodic Reset of Model Parameters.** During the training of our ZO Fine-tuner, a lot of data needs to be generated. However, since the optimizer is trained along the fine-tuning trajectory of a model using a first-order optimizer, the auxiliary network tends to receive inputs that are chronologically ordered. In particular, it will get more data from the low-loss region. As a result, it may lead to overfitting to the low-loss region of the parameters while learning the crucial high-loss region insufficiently. To address this issue, we introduce a periodic re-initialization mechanism. After each complete optimization cycle or when the loss has sufficiently decreased by the first-order guidance, we reset the model parameters to their original pre-finetuning state and restart the fine-tuning process. This approach introduces diversity into the input distribution by exposing the optimizer to model states from multiple phases of training.

**Complete Learning to Learn Framework for ZO Fine-tuner Training.** Algorithm 2 illustrates the complete L2L framework for training ZO Fine-tuner. At each training step, we sample a training dataset and a batch from this dataset to perform a one-step update to ZO Fine-tuner as described above. Moreover, we periodically reset the model parameters to mitigate the bias discussed previously. Despite the complexity of this training algorithm, we would like to emphasize that it is a *one-time* cost: once ZO Fine-tuner is learned, deployment reduces to Algorithm 1.

## 4. Experiment

Following MeZO (Malladi et al., 2023), we evaluate ZO Fine-tuner with four LLMs: LLaMA-3.2-1B (Grattafiori et al., 2024), LLaMA-3.1-8B (Grattafiori et al., 2024),

---

**Algorithm 2** Learning to Learn Framework

**Input:** LLM parameters $\theta$, training step $T$, loss function for LLM $\mathcal{L}$, learning rate for LLM $\eta_1$, learning rate for PertNN $\eta_2$, task list $\mathcal{T}_{\text{list}}$, perturbation scale $\epsilon$.
Initialize PertNN as $\omega_0$, LLM parameter as $\theta_0$, perturbation variance $\Sigma_0^{\mathcal{T}}$ as $I_d$ for each task $\mathcal{T}$.
**for** $t = 1, ..., T$ **do**
  $\mathcal{T}_{\text{list}} \leftarrow \text{Shuffle}(\mathcal{T}_{\text{list}})$
  **for** each task $\mathcal{T}$ in $\mathcal{T}_{\text{list}}$ **do**
    Sample a batch $\mathcal{B}_t^{\mathcal{T}}$ from $\mathcal{T}$
    $\Sigma_t^{\mathcal{T}} \leftarrow \text{PertNN}(\theta_t, \Sigma_{t-1}^{\mathcal{T}}, \boldsymbol{\ell}_{t-1}^{\mathcal{T}}; \omega_t)$
    Normalize such that $\|\Sigma_t^{\mathcal{T}}\|_F^2 = \|I_d\|_F^2$
    Sample $u_t \sim \mathcal{N}(0, \Sigma_t^{\mathcal{T}}(\Sigma_t^{\mathcal{T}})^\top)$ and compute LLM loss with perturbed parameter to obtain $\boldsymbol{\ell}_t$
    $\mathcal{L}_{\text{ZO}} \leftarrow \mathcal{L}^{\mathcal{T}}(\theta_t - \eta_1 \frac{l_t^+ - l_t^-}{2\epsilon} u_t; \mathcal{B}_t^{\mathcal{T}})$,
    $\omega_t \leftarrow \omega_t - \eta_2 \frac{\partial \mathcal{L}_{\text{ZO}}}{\partial \omega_t}$ {Update PertNN with SGD}
    $l_t \leftarrow \mathcal{L}^{\mathcal{T}}(\theta_t; \mathcal{B}_t^{\mathcal{T}})$
    $\theta_t \leftarrow \theta_t - \eta_1 \frac{\partial l_t}{\partial \theta_t}$     {Update LLM with SGD}
  **end for**
  $\omega_{t+1} \leftarrow \omega_t, \theta_{t+1} \leftarrow \theta_t$
  When the training step $t$ reaches a predefined period, reinitialize LLM parameter as $\theta_0$.
**end for**

---

Qwen2.5-14B (Bai et al., 2023), and OPT-30B (Zhang et al., 2022) using seven diverse benchmark datasets including SST-2 (Socher et al., 2013), CB (De Marneffe et al., 2019), COPA (Roemmele et al., 2011), BoolQ (Clark et al., 2019), WSC (Levesque et al., 2012), SQuAD (Rajpurkar et al., 2016), and DROP (Dua et al., 2019).

We compare our approach against four representative zeroth-order optimization baselines for LLM fine-tuning: HIZOO (Zhao et al., 2025), LOZO (Chen et al., 2024), MeZO and MeZO-Adam (Malladi et al., 2023). Due to computational resource constraints, we replace the expensive MeZO-Adam with a more efficient variant, MeZO-AdamU (Jiang et al., 2023), for models larger than LLaMA-3.2-1B. To ensure a fair comparison, we perform the same grid search over learning rates for each method and pick the best learning rate when reporting. Moreover, we used the same prompt templates as in the original MeZO for all baselines. More details can be found in Appendix B.2.

For our ZO Fine-tuner, we train it once using algorithm 2 on the COPA dataset. This choice is mainly due to COPA's consistently smooth loss decrease during standard fine-tuning, and its small size can enable fast training cycles. **Unless otherwise noted, the ZO Fine-tuner trained on COPA is reused as is across all other tasks and models in our main experiments.** Thus, all reported results provide a direct test of out-of-distribution transfer. In section 4.2, we also provide ablations regarding the choice of meta-learning

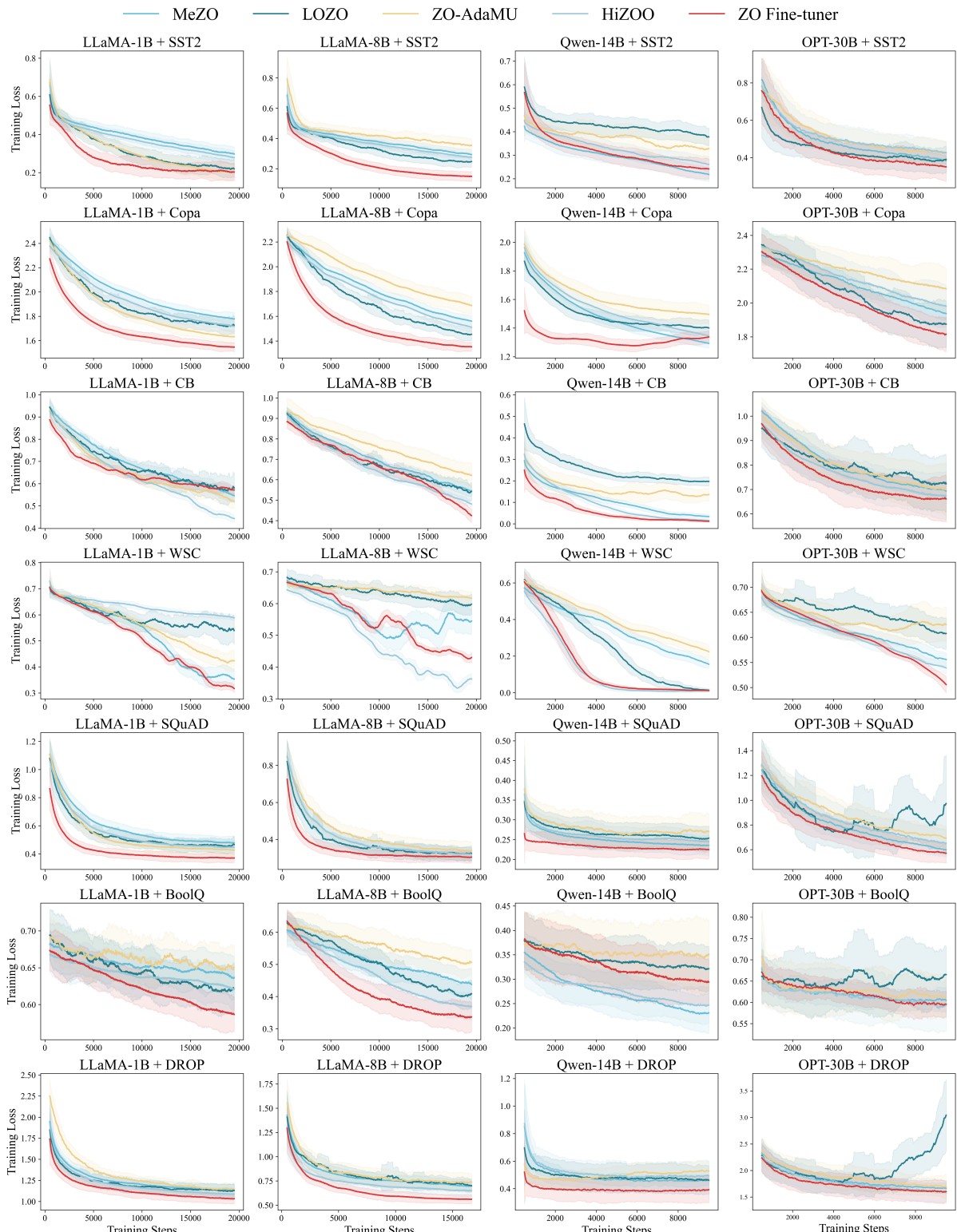

*Figure 2.* Loss comparison across different methods on various datasets and LLMs. Models (columns) are LLaMA-3.2-1B, LLaMA-3.1-8B, Qwen2.5-14B, and OPT-30B, while datasets (rows) cover COPA, SST-2, CB, SQuAD, WSC, BoolQ, and DROP. All curves use the best hyperparameters found for each method. The shaded region around each curve shows the standard deviation of the smoothed loss. The wider the shade, the larger the fluctuation. ZO Fine-tuner shows advantages in both convergence speed and final loss value across most settings. We additionally provide visualizations on the accuracy curve in appendix C.6

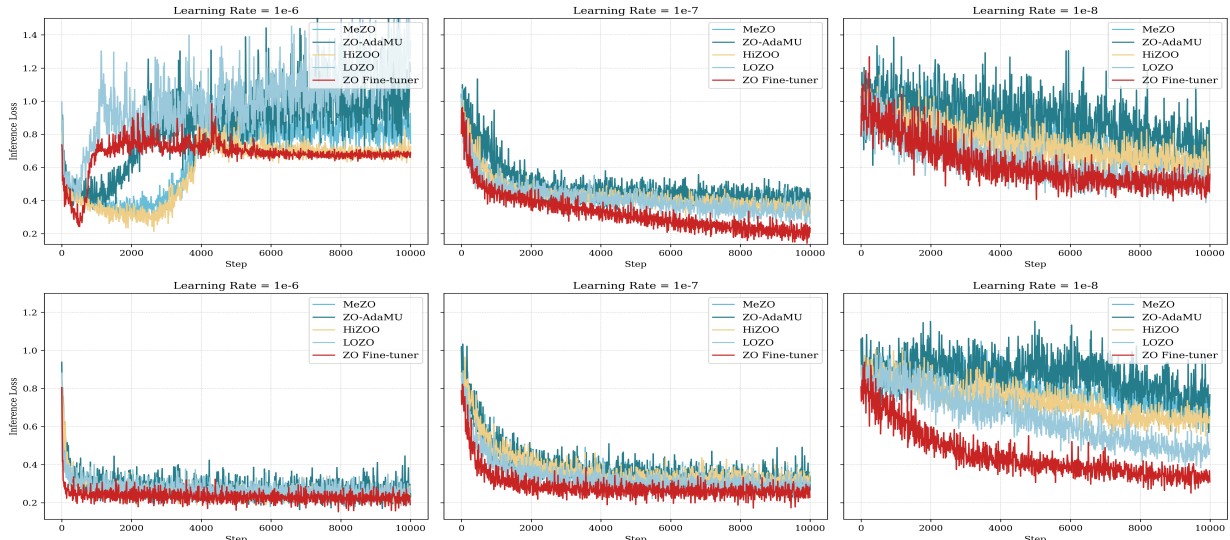

*Figure 3.* Loss curves under varying learning rates for different optimizers on (top) SST2 with LLaMA-3.1-8B, and (bottom) SQuAD with Qwen2.5-14B. It can be observed that ZO Fine-tuner is more robust to learning rate choices, and oftentimes achieves better convergence even under a smaller learning rate compared to the baselines.

dataset, and in Appendix C.5, we further discuss multi-dataset training. Other hyperparameters and training details can be found in section B.3 and section C.8.

### 4.1. Main Results

**Generalization Across Datasets.** Figure 2 compares convergence across all 28 dataset-model pairs using each method's best learning rate. We observe that ZO Fine-tuner (red) consistently reaches lower loss faster. The effect is especially clear on SST-2, CB, COPA, SQuAD, and DROP, where curves descend more steeply early on and settle at a better plateau. In addition, we report the final loss and accuracy values for all 28 combinations in Table 1. On average, ZO Fine-tuner achieves an average accuracy improvement of 2.5% over MeZO. Overall, our method outperforms the baselines in 75.0% of the task-model combinations in accuracy and 82.1% in the converged loss. These results indicate the strong generalization capability of ZO Fine-tuner, as training ZO Fine-tuner on a single COPA dataset already yields consistent gains across datasets. To show an upper bound of the improvements when full gradient is available, we provide the FO Adam baseline in appendix C.7.

**Generalization Across Model Derivatives.** We further evaluate whether ZO Fine-tuner transfers from a base model to its finetuned variants. Motivated by the fact that many checkpoints are obtained by further fine-tuning a small set of base LLMs, we train ZO Fine-tuner on LLaMA-3.1-8B and then apply it as is to fine-tune the derived checkpoint Llama-3.1-8B-Instruct. As shown in Table 4, ZO Fine-tuner consistently outperforms MeZO on both evaluated datasets in terms of average training loss and final accuracy.

racy. Practically, this supports a train once, reuse across derived checkpoints workflow: model developers could ship a pretrained finetuner with each base model, enabling downstream users to efficiently fine-tune derivative checkpoints with MeZO-style near-inference memory.

**Generalization to Longer-Sequence Datasets.** To assess whether our method generalizes to more commonly used longer sequence datasets, we also conduct experiments on a math-finetuning task, which has a long solution and is closer to modern reasoning training. Concretely, we fine-tuned Qwen2.5-14B on MetaMathQA (Yu et al., 2024) using the ZO Fine-tuner trained on COPA for 10000 steps, and compared it with MeZO under the same setup. The results can be found in Table 5, where we report the evaluation results on GSM8K (Cobbe et al., 2021) and Math-500 (Lightman et al., 2023). We observe that the COPA-trained ZO Fine-tuner still outperforms MeZO on this math dataset. This result suggests that the learned optimizer is not restricted to the GLUE/SuperGLUE-style tasks, and can transfer non-trivially to a substantially different task suite. This further validates our train once, reuse widely slogan.

### 4.2. Ablation Studies

**Learning Rate.** Figure 3 further demonstrates the sensitivity of different methods to the choice of learning rate. Notably, ZO Fine-tuner often achieves comparable loss at a learning rate of $1 \times 10^{-8}$ to that of baseline methods operating at $1 \times 10^{-7}$. When the learning rate further increases to $1 \times 10^{-6}$, many baseline methods suffer from instability and fail to converge, falling short of the performance that ZO Fine-tuner achieves at $1 \times 10^{-7}$.

*Table 1.* Average training loss in the final epoch and accuracy on seven datasets for each method and model combination under the best hyperparameter. We report both loss (↓) and accuracy / F1 (↑) across tasks of diverse formats to evaluate the overall performance.

| Model | Method | COPA | | SST-2 | | CB | | SQuAD | | WSC | | BoolQ | | DROP | |
|---|---|---|---|---|---|---|---|---|---|---|---|---|---|---|---|
| | | Loss | Acc | Loss | Acc | Loss | Acc | Loss | F1 | Loss | Acc | Loss | Acc | Loss | F1 |
| LLaMA-3.2-1B | MeZO | 1.77 | 0.75 | 0.29 | 0.90 | 0.55 | 0.70 | 0.48 | 0.75 | 0.35 | **0.62** | 0.63 | 0.63 | 1.16 | 0.29 |
| | MeZO-Adam | 1.62 | 0.79 | 0.20 | 0.92 | 0.53 | 0.66 | 0.41 | **0.78** | 0.42 | 0.61 | 0.66 | 0.62 | 1.14 | 0.29 |
| | HIZOO | 1.71 | 0.78 | 0.27 | 0.90 | **0.44** | **0.71** | 0.43 | 0.75 | 0.55 | 0.54 | 0.62 | 0.61 | 1.09 | 0.29 |
| | LOZO | 1.72 | 0.74 | 0.20 | 0.92 | 0.58 | 0.64 | 0.47 | **0.78** | 0.51 | 0.61 | 0.62 | 0.64 | 1.15 | 0.32 |
| | ZO Fine-tuner | **1.54** | **0.80** | **0.14** | **0.93** | 0.57 | 0.67 | **0.37** | **0.78** | **0.31** | 0.56 | **0.58** | **0.66** | **1.03** | **0.35** |
| LLaMA-3.1-8B | MeZO | 1.54 | 0.92 | 0.29 | 0.92 | 0.54 | 0.71 | 0.32 | 0.89 | 0.55 | 0.63 | 0.42 | 0.78 | 0.69 | 0.64 |
| | MeZO-AdamU | 1.67 | 0.89 | 0.36 | 0.92 | 0.61 | 0.70 | 0.35 | 0.86 | 0.61 | **0.64** | 0.50 | 0.75 | 0.73 | 0.59 |
| | HIZOO | 1.50 | **0.93** | 0.27 | 0.92 | 0.47 | 0.71 | 0.32 | 0.88 | **0.36** | 0.62 | 0.36 | 0.79 | 0.64 | 0.60 |
| | LOZO | 1.46 | 0.89 | 0.25 | **0.94** | 0.54 | 0.70 | 0.33 | **0.90** | 0.61 | 0.63 | 0.41 | 0.83 | 0.74 | 0.65 |
| | ZO Fine-tuner | **1.35** | 0.91 | **0.18** | **0.94** | **0.26** | **0.76** | **0.31** | **0.90** | 0.44 | 0.62 | **0.34** | **0.87** | **0.54** | **0.66** |
| Qwen2.5-14B | MeZO | 1.28 | 0.86 | **0.21** | 0.88 | 0.05 | **0.93** | 0.24 | 0.88 | 0.18 | 0.76 | **0.23** | 0.84 | 0.45 | 0.66 |
| | MeZO-AdamU | 1.43 | 0.85 | 0.35 | 0.89 | 0.13 | 0.91 | 0.28 | 0.90 | 0.25 | 0.75 | 0.35 | 0.84 | 0.50 | 0.64 |
| | HIZOO | **1.34** | 0.87 | 0.26 | 0.93 | **0.03** | 0.89 | 0.24 | 0.89 | **0.02** | **0.79** | 0.25 | 0.86 | 0.49 | 0.68 |
| | LOZO | 1.40 | 0.91 | 0.38 | 0.93 | 0.19 | 0.91 | 0.26 | 0.90 | 0.04 | **0.79** | 0.32 | 0.86 | 0.46 | 0.67 |
| | ZO Fine-tuner | **1.34** | **0.92** | 0.24 | **0.94** | 0.03 | **0.93** | 0.22 | **0.91** | 0.02 | 0.76 | 0.29 | **0.89** | **0.40** | **0.70** |
| OPT-30B | MeZO | 1.93 | 0.83 | 0.38 | 0.89 | 0.69 | 0.64 | 0.59 | 0.74 | 0.55 | **0.63** | **0.60** | 0.66 | 1.66 | **0.31** |
| | MeZO-AdamU | 2.07 | 0.80 | 0.43 | 0.84 | 0.70 | 0.66 | 0.67 | 0.73 | 0.62 | **0.63** | 0.62 | 0.66 | 1.70 | 0.30 |
| | HIZOO | 1.97 | 0.81 | 0.43 | 0.86 | 0.67 | 0.66 | 0.65 | 0.75 | 0.53 | 0.61 | 0.62 | 0.65 | 1.61 | 0.30 |
| | LOZO | 1.86 | 0.82 | 0.40 | **0.90** | 0.73 | 0.64 | 0.96 | 0.75 | 0.58 | 0.62 | 0.70 | 0.66 | 2.83 | 0.27 |
| | ZO Fine-tuner | **1.81** | **0.85** | 0.35 | 0.87 | **0.66** | **0.70** | **0.56** | **0.77** | 0.51 | 0.60 | 0.61 | **0.67** | **1.59** | **0.31** |

*Table 2.* Ablation results on Normalization and Periodic Reset. We report the final loss and the final accuracy. Consistently, both techniques individually improve performance across models and datasets, and combining them achieves the best results.

| Setting | LLaMA-8B + SQuAD | LLaMA-8B + SST2 | Qwen-14B + SQuAD | Qwen-14B + SST2 |
|---|---|---|---|---|
| Base | 0.3950 / 0.840 | 0.3976 / 0.874 | 0.3582 / 0.844 | 0.4086 / 0.800 |
| Reset alone | 0.3682 / 0.856 | 0.3891 / 0.881 | 0.3551 / 0.851 | 0.4039 / 0.810 |
| Normalize alone | 0.3071 / 0.899 | 0.3061 / 0.920 | 0.2380 / 0.904 | 0.3885 / 0.844 |
| Reset+Normalize | **0.3065 / 0.905** | **0.1789 / 0.941** | **0.2246 / 0.911** | **0.2403 / 0.935** |

*Table 3.* Ablation on parameter sharing strategy. We compare our block-wise scheme to a simpler layer-wise baseline. Block-wise sharing consistently achieves lower final loss and higher accuracy.

| Model | Sharing | SST2 Loss / Acc | SQuAD Loss / Acc |
|---|---|---|---|
| LLaMA-8B | layer-wise | 0.23 / 0.92 | 0.32 / 0.88 |
| | block-wise | **0.18 / 0.94** | **0.31 / 0.90** |
| Qwen-14B | layer-wise | 0.27 / 0.91 | 0.25 / 0.88 |
| | block-wise | **0.24 / 0.94** | **0.22 / 0.91** |

*Table 4.* We demonstrate that ZO Fine-tuner trained from LLaMA-3.1-8B generalizes well to LLaMA-3.1-8B-Instruct. Across datasets, it outperforms MeZO in final loss and accuracy.

| Method | Dataset | Loss / Acc |
|---|---|---|
| SST2 | MeZO | 0.276 / 0.92 |
| | ZO Fine-tuner | **0.164 / 0.95** |
| SQuAD | MeZO | 0.291 / 0.90 |
| | ZO Fine-tuner | **0.287 / 0.92** |

*Table 5.* Final test accuracy on mathematical reasoning datasets using Qwen2.5-14B finetuned on MetaMathQA using MeZO and ZO Fine-tuner.

| Dataset | Base Model | MeZO | ZO Fine-tuner |
|---|---|---|---|
| GSM8K | 78.9 | 81.4 | **85.6** |
| MATH-500 | 43.2 | 53.0 | **54.6** |

**Normalization & Periodic Reset.** We also conduct experiments to evaluate the effectiveness of our design choices, including normalization introduced in section 3.2 and periodic reset in section 3.3. From table 2, it is clear that both normalization and periodic reset help ZO Fine-tuner for achieving better performance. Moreover, combining the two strategies yields the best overall loss and accuracy.

**Parameter Sharing Strategy.** We also evaluate the granularity of sharing in ZO Fine-tuner, comparing our block-wise scheme to a simpler layer-wise sharing baseline. As shown in Table 3, block-wise sharing consistently achieves lower final loss and higher accuracy. Importantly, this choice is theory-driven: when the Hessian is (approximately) block-diagonal, theorem 3.1 indicates that the natural unit for variance sharing is the Hessian block itself.

**Meta-Learning Dataset.** Finally, we evaluate the robustness of ZO Fine-tuner under different meta-learning datasets. Specifically, we trained additional ZO Fine-tuners for LLaMA-3.2-1B using SQuAD as the meta-training dataset and for LLaMA-3.1-8B using SST-2 as the meta-training dataset. We then applied these learned optimizers to fine-tune the corresponding models on SST-2, COPA, and SQuAD. The results can be found in Table 6. It can be observed that the trained ZO Fine-tuner remains consistently better than MeZO. This demonstrates that the performance of ZO Fine-tuner is not attributed to some special properties of the COPA dataset, but rather our effective designs.

### 4.3. Memory Usage and Time Efficiency Analysis

**Memory Usage.** The memory overhead of ZO Fine-tuner when using to finetune LLMs mainly comes from the addi-

*Table 6.* Ablation on the meta-learning dataset. We report the final finetuning loss for ZO Fine-tuner compared with MeZO. It can be observed that ZO Fine-tuner trained on different datasets than COPA can still achieve superior performance compared to MeZO.

| Meta-training setting | Dataset | MeZO | ZO Fine-tuner |
|---|---|---|---|
| LLaMA-3.2-1B, SQuAD | SST-2 | 0.3095 | **0.2713** |
| | COPA | 1.7948 | **1.7604** |
| | SQuAD | **0.4827** | 0.4903 |
| LLaMA-3.1-8B, SST-2 | COPA | 1.5877 | **1.3669** |
| | SST-2 | 0.3060 | **0.1862** |
| | SQuAD | 0.3242 | **0.3076** |

tional memory taken by PertNN. However, the parameter number of our ZO Fine-tuner is extremely small, even negligible, compared to the LLMs. Consequently, the memory footprint of our method remains essentially identical to that of MeZO. Under equivalent experimental settings, it requires only $1/4$ of the memory overhead incurred by Adam. For example, MeZO and ZO Fine-tuner peak at 61GB and 62GB of GPU memory when fine-tuning OPT-30B, whereas first-order Adam reaches 312GB with FP16. More details can be found in Appendix C.3.

**Time Efficiency.** Similarly, the time overhead comes from the query to PertNN. However, this overhead is typically minimal. For example, when fine-tuning on DROP using LLaMA-3.2-1B on an L40S GPU with a batch size of 16, the generation of perturbation takes only 0.025 seconds, while all other operations take approximately 0.70 seconds. This means our method introduces less than 3.4% additional overhead, demonstrating time efficiency. This overhead becomes even less significant with larger models. For instance, under the same setting as LLaMA-3.1-8B, perturbation generation takes only 0.052 seconds compared to a total runtime of 3.14 seconds. More details can be found in Appendix C.2.

## 5. Conclusion

We introduced ZO Fine-tuner, a learned zeroth-order optimizer that learns adaptive per-block perturbation variances for efficient LLM fine-tuning. Experiments across four models and seven datasets show that a finetuner trained once on a single dataset transfers reliably to new tasks, finetuned derivative checkpoints, and broader task suites, supporting a practical "train once, reuse widely" workflow.

Our work focuses on improving optimization quality within the canonical MeZO-style zeroth-order regime. However, we note that techniques such as LoRA, quantization, and offloading are also important for reducing the memory footprint. Although not straightforward, we believe these methods can be potentially combined with ZO Fine-tuner to improve the memory-performance tradeoff in practice further. Systematically considering and evaluating these combinations is an important direction for future work.

## Acknowledgments

This work used the Delta system at the National Center for Supercomputing Applications (award OAC 2005572) through allocation CIS250462 from the Advanced Cyberinfrastructure Coordination Ecosystem: Services & Support (ACCESS) program, which is supported by National Science Foundation grants #2138259, #2138286, #2138307, #2137603, and #2138296.

## Impact Statement

This paper presents work whose goal is to advance the field of machine learning. In particular, our work aims to improve the practicality of LLM fine-tuning, which could have both beneficial and potentially harmful downstream uses. Other than this, there are many potential societal consequences of our work, none of which we feel must be specifically highlighted here."

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

# A. Discussions

## A.1. Limitations and Future Work

The coordinate-wise parameterization has been shown to be effective for learning-to-learn (L2L) zeroth-order optimization on small-scale models (Ruan et al., 2020). While our current design adopts a diagonal variance matrix $\Sigma_t$ for memory efficiency and strong empirical performance, exploring richer (e.g., non-diagonal) covariance structures is a promising direction. Such extensions could capture cross-parameter correlations, but would likely require additional techniques (e.g., structured parameterizations or low-rank factorizations) to control the associated memory and compute overhead.

More broadly, additional properties of LLM gradients and curvature may be exploitable. For example, prior work explicitly leverages low-rank structure in LLM gradients (Chen et al., 2024; Sun et al., 2025). An interesting future direction is to incorporate such structure into the perturbation design—either to generate more informative perturbations or to further reduce fine-tuning cost.

Finally, our experiments focus on improving optimization quality within the canonical MeZO-style zeroth-order regime. Techniques such as LoRA, quantization, and offloading are largely orthogonal system- or parameterization-level choices, and we expect them to be complementary to ZO Fine-tuner. Systematically evaluating and optimizing these combinations is an important direction for future work.

# B. Implementation Details

## B.1. Datasets and Models

We evaluate all optimizers on seven NLP tasks spanning multiple formats, including natural language inference, question answering, and commonsense reasoning. SST-2 (Socher et al., 2013) is a binary sentiment classification benchmark from the GLUE suite. CB (De Marneffe et al., 2019) and COPA (Roemmele et al., 2011) are low-resource natural language inference tasks from SuperGLUE, requiring models to recognize textual entailment or choose causal relationships. BoolQ (Clark et al., 2019) involves answering yes/no questions given short passages. WSC (Levesque et al., 2012) tests pronoun resolution in challenging coreference contexts. SQuAD (Rajpurkar et al., 2016) and DROP (Dua et al., 2019) are span-based question answering datasets that require locating answer spans in context paragraphs. For most classification tasks, we report accuracy as the evaluation metric. For SQuAD and DROP, we follow standard practice and report F1 score to better capture partial match quality.

We evaluate our optimizers on four representative large language models with diverse architectures and scales: LLaMA-3.2-1B (Grattafiori et al., 2024), LLaMA-3.1-8B (Grattafiori et al., 2024), Qwen2.5-14B (Bai et al., 2023), and OPT-30B (Zhang et al., 2022).

## B.2. Hyperparameters

We use a two-layer MLP with 64 hidden units and a tanh activation function as the auxiliary neural network for each parameter block. Table 7 presents the hyperparameter search grids used in our experiments to facilitate reproducibility. We primarily perform a grid search over three learning rate values: $10^{-4}$, $10^{-5}$, and $10^{-6}$ for MeZO-Adam, and $10^{-6}$, $10^{-7}$, and $10^{-8}$ for all other methods. For the WSC task, we additionally include $3 \times 10^{-7}$, as most methods exhibit slow loss decay at $10^{-7}$ and become unstable when using $10^{-6}$. We run 20,000 optimization steps on LLaMA-1B and LLaMA-8B, and 10,000 steps on Qwen-14B and OPT-30B due to resource limitation. A batch size of 16 is used for all models by default, except for OPT-30B, where we reduce it to 4 due to GPU memory constraints. Also due to computational resource constraints, we replace the expensive MeZO-Adam with its more efficient variant MeZO-AdamU (Jiang et al., 2023) for models larger than LLaMA-1B. As shown in the hyperparameter table, our method, together with MeZO, requires the smallest number of tunable hyperparameters among all baselines.

## B.3. Learning to Learn Details

In Section 3.3, we introduced our learning to learn framework. Here, we elaborate on additional implementation details. We use a two-layer MLP with 64 hidden units and a tanh activation function as the auxiliary neural network for each parameter block. Empirically, we set $\epsilon = 10^{-3}$, $\eta_1 = 10^{-6}$, and $\eta_2 = 10^{-2}$ in Algorithm 2. When the task list $\mathcal{T}_{\text{list}}$ contains only a single task, the framework reduces to single-dataset training as a special case. We find that training on a single dataset can

*Table 7.* Hyperparameter configurations for ZO Fine-tuner and all baseline methods.

| Method | Hyperparameters | Values |
|---|---|---|
| MeZO | Batch size | 16 for LLaMA-1B/8B/Qwen-14B; 4 for OPT-30B |
| | Learning rate | $\{10^{-6}, 10^{-7}, 10^{-8}\}$ (plus $3 \times 10^{-7}$ for WSC only) |
| | $\epsilon$ | $10^{-3}$ |
| MeZO-Adam | Batch size | 16 |
| | Learning rate | $\{10^{-4}, 10^{-5}, 10^{-6}\}$ (plus $3 \times 10^{-6}$ for WSC only) |
| | $\epsilon$ | $10^{-3}$ |
| | $\epsilon_{\text{Adam}}$ | $\{10^{-6}, 10^{-7}, 10^{-8}\}$ |
| ZO-AdamU | Batch Size | 16 for LLaMA-1B/8B/Qwen-14B; 4 for OPT-30B |
| | Learning Rate | $\{10^{-6}, 10^{-7}, 10^{-8}\}$ (plus $3 \times 10^{-7}$ for WSC only) |
| | $\epsilon$ | $10^{-3}$ |
| | $\alpha$ | $\{0.2, 0.5, 0.7\}$ |
| | $\beta^{(1)}$ | $\{0.9, 0.8, 0.7\}$ |
| | $\beta^{(2)}$ | $\{0.01, 0.05, 0.1\}$ |
| HIZOO | Batch Size | 16 for LLaMA-1B/8B/Qwen-14B; 4 for OPT-30B |
| | Learning Rate | $\{10^{-6}, 10^{-7}, 10^{-8}\}$ (plus $3 \times 10^{-7}$ for WSC only) |
| | $\epsilon$ | $10^{-3}$ |
| | Smooth Constant | $\{10^{-7}, 10^{-8}\}$ |
| LOZO | Batch Size | 16 for LLaMA-1B/8B/Qwen-14B; 4 for OPT-30B |
| | Learning Rate | $\{10^{-6}, 10^{-7}, 10^{-8}\}$ (plus $3 \times 10^{-7}$ for WSC only) |
| | $\epsilon$ | $10^{-3}$ |
| | Rank ($r$) | $\{2, 4\}$ |
| | Interval ($\nu$) | $\{50, 100\}$ |
| ZO Fine-Tuner | Batch Size | 16 for LLaMA-1B/8B/Qwen-14B; 4 for OPT-30B |
| | Learning Rate | $\{10^{-6}, 10^{-7}, 10^{-8}\}$ (plus $3 \times 10^{-7}$ for WSC only) |
| | $\epsilon$ | $10^{-3}$ |

yield competitive performance with reduced cost. In our experiments, the optimizer is trained on COPA. A comparison between single-dataset and multi-dataset training results is provided in Section C.5. We also block certain gradient flows to reduce memory consumption during learning-to-learn. Specifically, recall that

$$\hat{g}(\theta_t; \omega) = \frac{\mathcal{L}(\theta_t + \epsilon u_t) - \mathcal{L}(\theta_t - \epsilon u_t)}{2\epsilon} u_t,$$
$$u_t = \text{PertNN}(\theta_t, \Sigma_{t-1}, \boldsymbol{\ell}_{t-1}; \omega) z_t, z_t \sim \mathcal{N}(0, I_d).$$

The gradient of the ZO loss, defined as $\mathcal{L}_{\text{ZO}}(\theta; \omega) := \mathcal{L}(\theta - \eta \hat{g}(\theta; \omega))$, propagates first to $\hat{g}(\theta_t; \omega)$, and then further through both components used to construct it: the perturbation direction $u_t$ and the finite-difference estimator $\frac{\mathcal{L}(\theta_t + \epsilon u_t) - \mathcal{L}(\theta_t - \epsilon u_t)}{2\epsilon}$. To save memory, we cut off the gradient flow through the finite-difference term, which eliminates the need to back-propagate through the inner loss evaluations and store their activations. Despite this approximation, we still observe strong empirical performance.

# C. Additional Experimental Results

## C.1. Additional Results on Learning Rate Sensitivity

We previously presented the sensitivity of different optimization methods to the learning rate in Section 4.2. Due to space constraints, only a subset of the results was shown. Here, we provide the complete loss curves across the three benchmarks SQuAD, SST-2, and COPA using LLaMA-1B, LLaMA-8B, Qwen-14B, and OPT-30B, as shown in Figure 4 and Figure 5.

Across the grid search over learning rates $10^{-6}$, $10^{-7}$, and $10^{-8}$, the ZO Fine-tuner consistently achieves superior results compared to all baselines when comparing their best-performing settings. On LLaMA-1B, LLaMA-8B, Qwen-14B and OPT-30B, our method exhibits faster convergence and achieves lower final loss, particularly under the two learning rates $10^{-7}$ and $10^{-8}$. Notably, ZO Fine-tuner often matches or exceeds the best performance of other methods at $10^{-7}$, even when operating at $10^{-8}$ on LLaMA-1B, LLaMA-8B and Qwen-14B.

When increasing the learning rate from $10^{-8}$ to $10^{-7}$, ZO Fine-Tuner continues to improve. In contrast, baseline methods tend to suffer from instability at higher learning rates like $10^{-6}$ when increasing from $10^{-7}$, especially on LLaMA-1B and LLaMA-8B. At the low end ($10^{-8}$), many baselines exhibit stagnation, which means their loss decreases slowly or plateaus. This suggests limited adaptivity in low-gradient regimes. These observations underscore the robustness of ZO

*Table 8.* Component-wise runtime breakdown (in seconds and percentage of total time) for different models. All results are tested on DROP and L40S GPU with a batch size of 16 using FP16.

| Model | Generate Var | Perturb Param | Update Param | Compute Loss | Total Time |
|---|---|---|---|---|---|
| LLaMA-1B | 0.025s (3.39%) | 0.052s (7.07%) | 0.021s (2.86%) | 0.631s (86.65%) | 0.729s |
| LLaMA-8B | 0.052s (1.66%) | 0.460s (14.67%) | 0.192s (6.11%) | 2.433s (77.55%) | 3.137s |
| Qwen-14B | 0.119s (2.06%) | 0.395s (6.82%) | 0.164s (2.84%) | 5.106s (88.27%) | 5.785s |
| OPT-30B | 0.142s (1.64%) | 0.214s (2.48%) | 0.090s (1.04%) | 8.183s (94.82%) | 8.630s |

*Table 9.* Peak GPU memory usage (GB) of different optimization methods across models on the SST-2 dataset, using batch size = 1 and FP16 precision.

| Method | LLaMA-1B | LLaMA-8B | Qwen-14B | OPT-30B |
|---|---|---|---|---|
| MeZO | 5 | 20 | 35 | 61 |
| LOZO | 5 | 20 | 35 | 61 |
| HiZOO | 6 | 23 | 40 | 65 |
| ZO-AdaMU | 9 | 39 | 69 | 122 |
| ZO Fine-Tuner | 5 | 21 | 36 | 62 |
| FO-SGD | 9 | 40 | 74 | 126 |
| FO-Adam | 13 | 84 | 163 | 316 |

Fine-Tuner across a wide range of learning rates and tasks, highlighting its strong default behavior even without fine-tuned hyperparameters.

### C.2. Time Analysis

We further break down the runtime of each component involved in the optimizer and summarize the results in Table 8. Among these, the variance generation step is extremely lightweight. It only accounts for 3.39% of total runtime on LLaMA-1B, and less than 2.06% on larger models such as LLaMA-8B, Qwen-14B, and OPT-30B. This highlights the efficiency of our design: although we introduce an additional learned component to control perturbation variance, it imposes almost no computational overhead.

This can be easily explained by the fact that we only employ a lightweight neural network for each parameter block, specifically a two-layer MLP with just 32 hidden units. In addition, both the input and output of these networks are compressed, further reducing the computational cost.

In contrast, the dominant cost comes from loss computation, which includes forward passes for both positive and negative perturbations. This accounts for over 77%–95% of total runtime and is intrinsic to all zeroth-order optimization frameworks. Overall, our method introduces minimal additional cost while achieving adaptive and effective optimization.

### C.3. Memory Analysis

Table 9 reports the peak GPU memory usage of various optimization methods across different model sizes on the SST-2 dataset. We observe that all zeroth-order (ZO) methods, including MeZO, LOZO, and HiZOO, exhibit similar memory footprints. The only notable exception is ZO-AdaMU, which incurs higher memory usage due to its additional momentum tracking. Compared to the first-order method like Adam, all ZO methods consume significantly less memory, highlighting the efficiency of ZO-based optimization. Notably, our ZO Fine-Tuner achieves comparable memory usage to other ZO baselines, indicating that it introduces no additional memory overhead beyond standard ZO designs.

### C.4. Cost of Learning to Learn

We also assess the time and memory overhead incurred during the training of the ZO Fine-Tuner in table 10. In general, L2L takes approximately 2.4× the time of standard first-order fine-tuning. This is expected, as the L2L process inherently includes a full fine-tuning phase using SGD. However, importantly, this cost is incurred only **once**, as a single ZO Fine-Tuner trained for a given model can be reused across diverse downstream tasks, effectively amortizing the training cost.

*Table 10.* Time and memory cost of meta-training the ZO Fine-Tuner in our L2L framework. GPU memory usage and GPU time (in minutes) are reported for different foundation models. This cost is incurred only once per base model, and the trained fine-tuner can be reused across downstream tasks.

| Model | GPU Memory (GB) | Meta-training GPU Time (min) |
|---|---|---|
| LLaMA-1B | 13 | 3 |
| LLaMA-8B | 83 | 15 |
| Qwen-14B | 150 | 25 |
| OPT-30B | 332 | 51 |

*Table 11.* FO reference results on LLaMA-3.2-1B across datasets. FO uses Adam fine-tuning and is reported for context; MeZO and ZO Fine-tuner operate in the MeZO-style zeroth-order regime.

| Setting | COPA | CB | SQuAD | WSC | BoolQ | DROP |
|---|---|---|---|---|---|---|
| MeZO | 0.75 | 0.70 | 0.75 | 0.62 | 0.63 | 0.29 |
| Ours | **0.80** | 0.73 | 0.80 | 0.57 | 0.66 | 0.32 |
| FO (Adam) | 0.76 | **0.75** | **0.82** | 0.60 | **0.69** | **0.45** |

### C.5. Comparison between Single-dataset and Multi-dataset Training

We also compare the performance of ZO Fine-tuner under single-dataset and multi-dataset training settings. In the multi-dataset setting, we construct a diverse training set by selecting one representative dataset from each task type: SST-2 for sentiment analysis, CB and COPA for natural language inference, and SQuAD for question answering. For the single-dataset setting, the optimizer is trained solely on COPA.

The multi-dataset setting could lead to better performance. However, as shown in Figure 7, in some cases, the ZO Fine-tuner trained on a single dataset can outperform its multi-dataset counterpart. Overall, the two settings yield comparable performance. Single-dataset training is also simpler to implement and tune, while still achieving competitive results. And that's why we choose to use it throughout the main experiments.

### C.6. Accuracy Curve

Although we believe that for our work, the loss curve is a more direct reflection of the optimizer's effectiveness, we also provide accuracy curves for comprehensiveness. In particular, we plotted the accuracy curve for ZO Fine-tunerand MeZO on SST2 and SQuAD dataset with LLaMA-3.1-8B. It is clear that our ZO Fine-tunerachieves faster convergence and higher final accuracy compared to MeZO.

### C.7. Comparison with First-Order Baselines

To contextualize the performance gap between zeroth-order (ZO) fine-tuning and gradient-based training, we report first-order (FO) fine-tuning results. FO baselines require backpropagation and optimizer states, and thus fall outside the MeZO-style ZO regime studied in the main paper; we include them here only to provide a practical upper bound on attainable accuracy when gradients are available.

Table 11 reports final accuracy across all datasets for LLaMA-3.2-1B when fine-tuned with Adam. Table 12 additionally reports SST-2 results across multiple models. Overall, FO methods generally outperform ZO methods, while ZO Fine-tuner remains competitive with a modest performance gap in most settings.

### C.8. Additional L2L Implementation Details

**Trajectory choices (SGD vs. Adam).** Our L2L training uses a first-order trajectory to generate model states for optimizer learning. We provide an ablation of the trajectory optimizer (SGD vs. Adam) on SST-2 for LLaMA-3.2-1B and LLaMA-3.1-8B in Table 13 and Table 14. Across intervals, Adam and SGD lead to similar loss curves and final test accuracy; in some intervals, SGD is slightly better. Since SGD has a substantially smaller memory footprint than Adam, we use SGD to generate trajectories for all reported L2L runs.

**Periodic reset schedule.** We tune the reset schedule based on the first-order trajectory behavior. For example, for LLaMA-

*Table 12.* SST-2 accuracy across models. FO is a gradient-based reference; MeZO and ZO Fine-tuner are zeroth-order methods.

| Setting | LLaMA-3.2-1B | LLaMA-3.1-8B | Qwen2.5-14B |
|---------|--------------|--------------|-------------|
| MeZO | 0.90 | 0.92 | 0.88 |
| Ours | 0.92 | **0.94** | **0.94** |
| FO | **0.93** | 0.93 | **0.94** |

*Table 13.* Ablation on first-order trajectory choice (SGD vs. Adam) for LLaMA-3.2-1B on SST-2. Values are average training loss over the specified step intervals.

| | Avg Loss [0,4k] | [4k,8k] | [8k,12k] | [12k,16k] | [16k,20k] |
|---|---|---|---|---|---|
| SGD | 0.3732 | 0.2219 | 0.1855 | 0.1670 | 0.1575 |
| Adam | 0.3539 | 0.2213 | 0.1962 | 0.1808 | 0.1655 |

3.2-1B on COPA, after 5 epochs of SGD the loss typically drops to around 0.001; we therefore reset every 5 epochs and train ZO Fine-tuner for 15 epochs total (with data reshuffling). We apply analogous tuning for other models.

**L2L learning rates.** The L2L pipeline involves multiple learning rates (for the first-order trajectory, the zeroth-order updates, and updating ZO Fine-tuner), making exhaustive sweeps impractical. Our choices (Appendix C.3) are guided by earlier empirical testing; we observe that larger learning rates than those reported frequently lead to instability or failure. We therefore fix these hyperparameters across experiments.

**One-time tuning and reproducibility.** Although tuning the L2L process introduces additional design choices, this tuning is performed only once per base model, and the resulting ZO Fine-tuner is reused across datasets and derivative checkpoints. We also release our trained finetuner checkpoints and fixed-seed scripts to facilitate reproducibility.

## D. Theoretical Analysis

In this section, we formally discuss our theoretical results and derive theorem 3.1. First, we set up the notations and definition we need and formally present the theorem.

**Definition D.1** (Expected Loss Change). The expected change in loss after performing a one-step update from parameter $\theta_t$ is defined as

$$d(\theta_t) := \mathbb{E}[\mathcal{L}(\theta_{t+1}) \mid \theta_t] - \mathcal{L}(\theta_t).$$

**Assumption D.2** (Local $r$-effective rank). Let $G(\theta_t) = \max_{(x,y)\in D} \|\nabla\mathcal{L}(\theta_t; \{(x,y)\})\|$. There exists a matrix $H(\theta_t) \le \ell \cdot I_d$ such that:

1. For all $\theta$ such that $\|\theta - \theta_t\| \le \eta dG(\theta_t)$, we have $\nabla^2\mathcal{L}(\theta) \preceq H(\theta_t)$.

2. The effective rank of $H(\theta_t)$, i.e. $\operatorname{tr}(H(\theta_t))/\|H(\theta_t)\|_{\mathrm{op}}$, is at most $r$.

**Theorem D.3.** *Under Assumption D.2, the expected loss change after one-step update of MeZO has upper bound as follows, where $\Sigma_{MB} = \operatorname{Cov}(\nabla\mathcal{L}(\theta_t; \{(x_i, y_i)\}))$:*

$$d_{\mathrm{MeZO}}(\theta_t) = \mathbb{E}[\mathcal{L}(\theta_{t+1})|\theta_t] - \mathcal{L}(\theta_t)$$
$$\le -\eta\|\nabla\mathcal{L}(\theta_t)\|^2 + \frac{\eta^2\ell}{2} \cdot \left(\frac{dr + d - 2}{d + 2} + 1\right)$$
$$\cdot \left(\|\nabla\mathcal{L}(\theta_t)\|^2 + \frac{1}{B}\operatorname{tr}(\Sigma_{MB}(\theta_t))\right)$$

**Assumption D.4** (Local Block-wise $r_i$-Effective Rank). The Hessian matrix $H(\theta_t)$ in Assumption D.2 satisfies the following property: $H(\theta_t) = \operatorname{diag}(H_1(\theta_t), \ldots, H_m(\theta_t))$ and $r_i := \operatorname{tr}(H_i(\theta_t))/\|H_i(\theta_t)\|_{\mathrm{op}}$ have different upper bounds $r_i$.

**Theorem D.5.** *Under Assumption D.2 and Assumption D.4 and ideal situation, assigning distinct perturbation variances across parameter blocks can yield a tighter upper bound than that of $d_{\mathrm{MeZO}}(\theta_t)$.*

*Table 14.* Ablation on first-order trajectory choice (SGD vs. Adam) for LLaMA-3.1-8B on SST-2. Values are average training loss over the specified step intervals.

| | Avg Loss [0,4k) | [4k,8k) | [8k,12k) | [12k,16k) | [16k,20k) |
|---|---|---|---|---|---|
| SGD | 0.4276 | 0.2758 | 0.2056 | 0.1702 | 0.1527 |
| Adam | 0.4159 | 0.2933 | 0.2303 | 0.1917 | 0.1686 |

Assumption D.2 and Theorem D.3 are directly from the theoretical analysis of MeZO (Malladi et al., 2023). MeZO states the Lipschitz condition alone does not guarantee convergence in high-dimensional settings and it is necessary to **leverage the low-rank structure of the Hessian matrix**. Assumption D.4 is consistent with the actual situation, which has been deeply researched and checked by work like Adam-mini (Zhang et al., 2024b;a).

In this section, we consider a setting where, at each iteration, we sample a perturbation block-by-block: for the $i$-th parameter block, we draw noise from the distribution $N(0, \sigma_i I_{d_i})$, apply the perturbation to the $i$-th block of parameters, and perform a zeroth-order update accordingly. Let the total number of blocks be $b$, and denote by $\mathcal{L}(\theta_{t,j})$ the loss after perturbing the $j$-th block at iteration $t$. We further denote the full loss after perturbing all $b$ blocks as $\mathcal{L}(\theta_{t+1})$, and let $\nabla_j \mathcal{L}(\theta_{t,j})$ denote the $j$-th block of the gradient evaluated at $\theta_{t,j}$. Here, we adopt the sphere (normalized-Gaussian) perturbation used in the original MeZO analysis for its built-in step-size control. An analogous convergence form also holds for Gaussian perturbations, as shown in prior work(Malladi et al., 2023), when the probability of large updates $\|\theta_{t+1} - \theta_t\|$ is kept small, which ensures the required local assumptions hold with high probability.

*Proof.* As shown in Theorem D.3, the expected loss decrease under MeZO is bounded by

$$d_{\text{MeZO}}(\theta_t) = \mathbb{E}[\mathcal{L}(\theta_{t+1})|\theta_t] - \mathcal{L}(\theta_t)$$

$$\leq -\eta\|\nabla\mathcal{L}(\theta_t)\|^2 + \frac{\eta^2\ell}{2} \cdot \left(\frac{dr + d - 2}{d + 2} + 1\right)$$

$$\cdot \left(\|\nabla\mathcal{L}(\theta_t)\|^2 + \frac{1}{B}\text{tr}(\Sigma_{MB}(\theta_t))\right)$$

Since each block-wise gradient estimate is still an unbiased estimator of the true gradient restricted to the corresponding block, we can get:

$$\mathbb{E}[\mathcal{L}(\theta_{t,j+1})|\theta_{t,j}] - \mathcal{L}(\theta_{t,j}) \leq$$

$$-\eta\sigma_j^2\|\nabla_j\mathcal{L}(\theta_{t,j})\|^2 + \frac{\eta^2\sigma_j^4\ell}{2} \cdot \left(\frac{dr_j + d - 2}{d + 2} + 1\right)$$

$$\cdot \left(\|\nabla_j\mathcal{L}(\theta_{t,j})\|^2 + \frac{1}{B}\text{tr}(\Sigma_{MB,j}(\theta_{t,j}))\right)$$

By summing both sides over $j = 1$ to $b$ and taking expectation, we eliminate the dependence on $\theta_{t,j}$: the right-hand side becomes a function of $\theta_t$ only, while the left-hand side depends only on $\theta_{t+1}$ and $\theta_t$. This yields:

$$\mathbb{E}[\mathcal{L}(\theta_{t+1})|\theta_t] - \mathcal{L}(\theta_t) \leq -\eta\sum_{j=1}^{b}\sigma_j^2\mathbb{E}[\|\nabla_j\mathcal{L}(\theta_{t,j})\|^2|\theta_t]$$

$$+ \sum_{j=1}^{b}\frac{\eta^2\sigma_j^4\ell}{2} \cdot \left(\frac{dr_j + d - 2}{d + 2} + 1\right)$$

$$\cdot \left(\mathbb{E}[\|\nabla_j\mathcal{L}(\theta_{t,j})\|^2|\theta_t] + \frac{1}{B}\text{tr}(\mathbb{E}[\Sigma_{MB,j}(\theta_{t,j})|\theta_t])\right)$$

$$= \sum_{j=1}^{b} -\eta\sigma_j^2\|\nabla_j\mathcal{L}(\theta_t)\|^2 + \sum_{j=1}^{b}\frac{\eta^2\sigma_j^4\ell}{2} \cdot$$

$$\left(\frac{dr_j + d - 2}{d + 2} + 1\right) \cdot \left(\|\nabla_j\mathcal{L}(\theta_t)\|^2 + \frac{1}{B}\text{tr}(\Sigma_{MB}(\theta_t))\right)$$

The equality in the last line follows from the condition that the Hessian matrix is block-diagonal according to Assumption D.4. Specifically, when updating block $j_1$, the change in the gradient of block $j_2$ ($j_2 \neq j_1$) can be expressed as:

$$\nabla_{j_2}\mathcal{L}(\theta_{t,j_1}) - \nabla_{j_2}\mathcal{L}(\theta_t) = \int_0^1 H_{j_2 j_1}(\theta_t + sP_{j_1}\delta)\,\delta\,ds,$$

where $P_{j_1}$ denotes the projection onto block $j_1$, and $H_{j_2,j_1}(\cdot)$ is the $(j_2, j_1)$ block of the Hessian. The perturbation direction $\delta$ is sampled from the standard multivariate normal distribution and scaled by the corresponding block-wise variance, i.e., $\delta \sim \mathcal{N}(0, \sigma_{j_1}^2 I_{d_{j_1}})$ for block $j_1$. Under the block-diagonal assumption, $H_{j_2,j_1}(\cdot) = 0$ for all $j_2 \neq j_1$, and thus cross-block gradient changes vanish.

Even if Assumption D.4 does not hold exactly, the effect of cross-block interactions can still be controlled by bounding the operator norm of the off-diagonal blocks of the Hessian. As long as these terms remain small, the overall error introduced in the bound remains negligible.

Note that if we set $\sigma_j = 1$ for all $j$, our upper bound reduces to the standard MeZO bound:

$$
\begin{aligned}
&\mathbb{E}[\mathcal{L}(\theta_{t+1})|\theta_t] - \mathcal{L}(\theta_t) \\
&\leq \sum_{j=1}^{b} -\eta\|\nabla_j\mathcal{L}(\theta_t)\|^2 + \sum_{j=1}^{b} \frac{\eta^2\ell}{2}\cdot\left(\frac{dr_j + d - 2}{d+2} + 1\right) \\
&\quad \cdot\left(\|\nabla_j\mathcal{L}(\theta_t)\|^2 + \frac{1}{B}\mathrm{tr}(\Sigma_{MB}(\theta_t))\right) \\
&\leq -\eta\|\nabla\mathcal{L}(\theta_t)\|^2 + \frac{\eta^2\ell}{2}\cdot\left(\frac{dr + d - 2}{d+2} + 1\right) \\
&\quad \cdot\left(\|\nabla\mathcal{L}(\theta_t)\|^2 + \frac{1}{B}\mathrm{tr}(\Sigma_{MB}(\theta_t))\right)
\end{aligned}
$$

where $r$ is the (uniform) effective rank used in MeZO and $r \geq r_j$ for any $j$. Therefore, by optimizing $\sigma_j$ for each block according to its local structure (e.g., $r_j$), we can obtain a strictly tighter upper bound than $d_{\mathrm{MeZO}}(\theta_t)$. $\qquad\square$

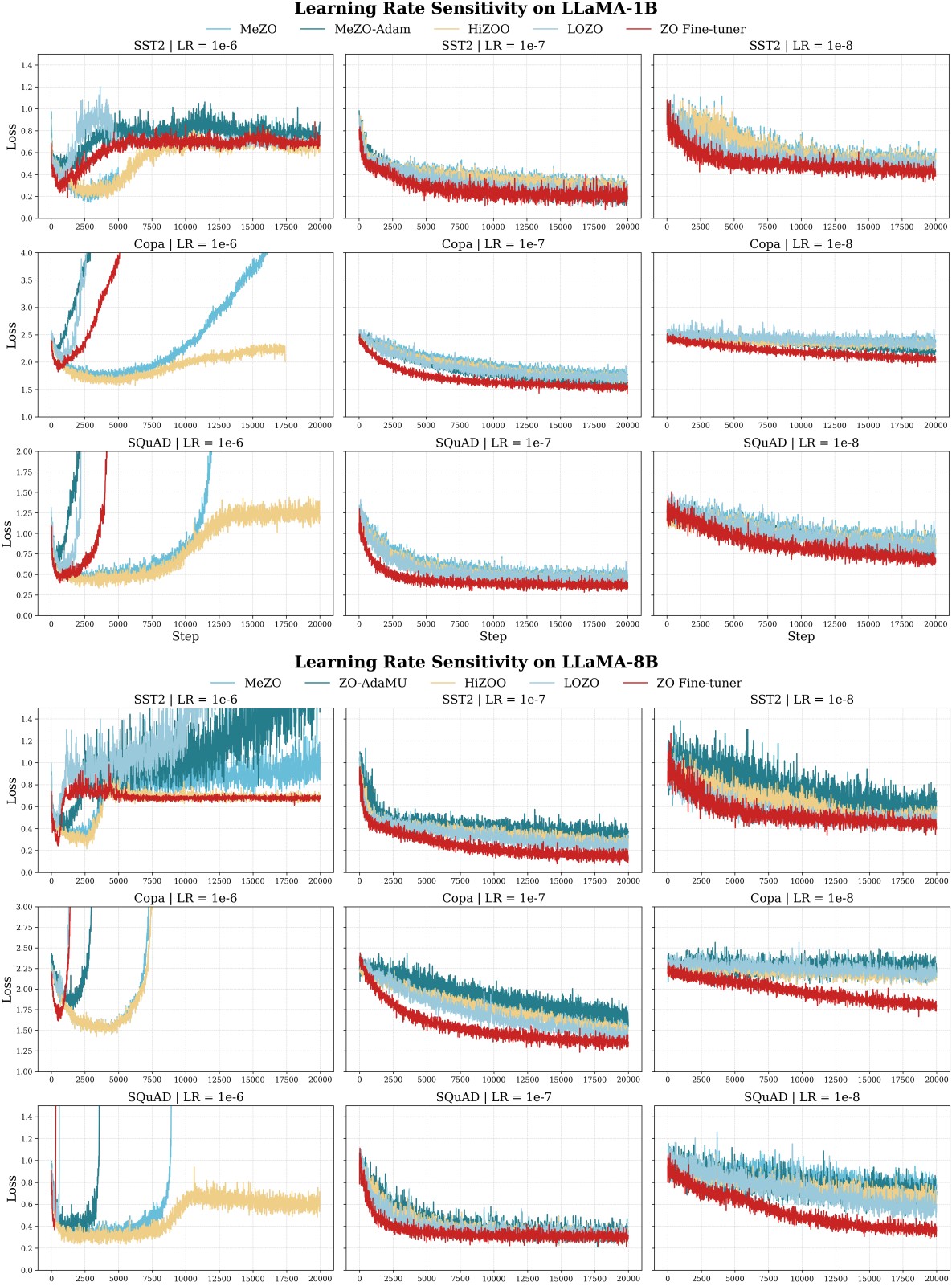

*Figure 4.* Loss curves under varying learning rates for different optimizers with LLaMA-1B (top) and Qwen-14B (bottom). We report results on SST2, Copa, and SQuAD. For MeZO-Adam, note that the actual learning rates used were $10^{-4}$, $10^{-5}$, and $10^{-6}$, corresponding to the plotted values of $10^{-6}$, $10^{-7}$, and $10^{-8}$, respectively.

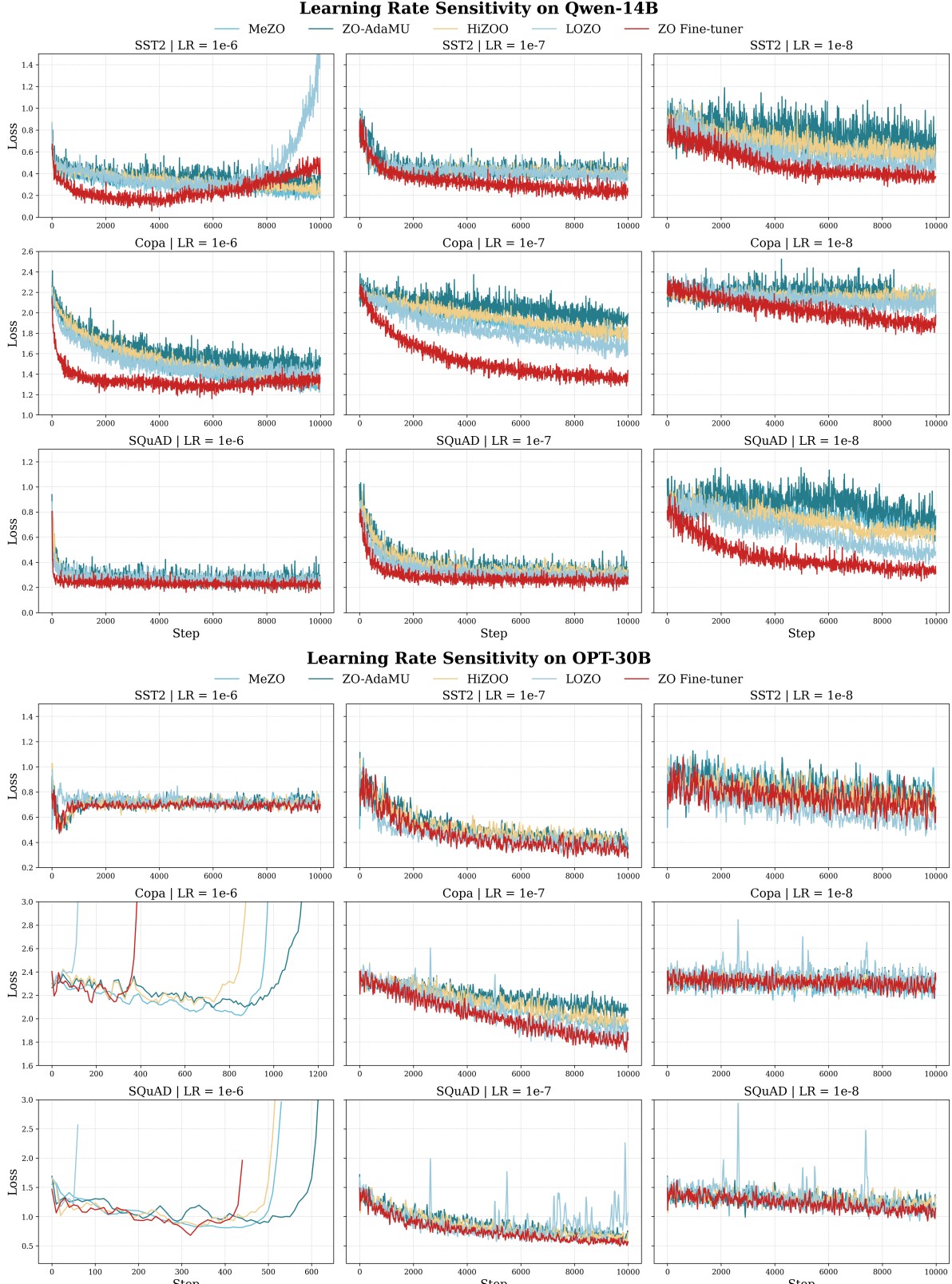

*Figure 5.* Loss curves under varying learning rates for different optimizers with Qwen-14B (top) and OPT-30B (bottom). We report results on SST2, Copa, and SQuAD.

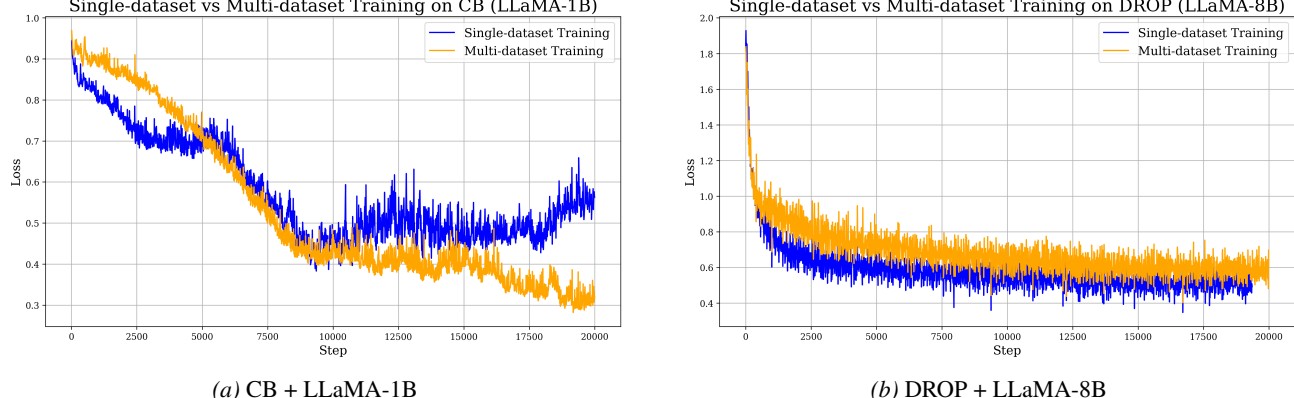

*(a)* CB + LLaMA-1B                    *(b)* DROP + LLaMA-8B

*Figure 6.* Comparison of inference loss between ZO Fine-tuners trained with single-dataset and multi-dataset settings. Results are reported on CB task using LLaMA-1B (left) and on DROP task LLaMA-8B (right). The single-dataset variant is trained solely on COPA, while the multi-dataset variant is jointly trained on COPA, SST-2, and SQuAD.

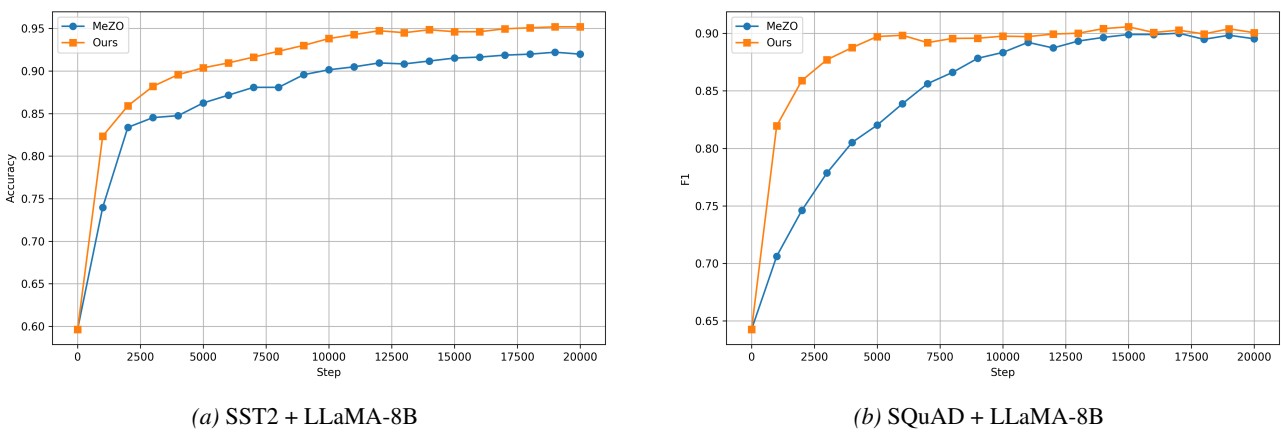

*(a)* SST2 + LLaMA-8B                    *(b)* SQuAD + LLaMA-8B

*Figure 7.* Test accuracy for ZO Finetuner and MeZO on SST2 and SQuAD with LLaMA-3.1-8B.

