# OpenReview forum: "Learning a Zeroth-Order Optimizer for Fine-Tuning LLMs"
_ICML.cc/2026/Conference — ICML 2026 regular_

### Official Review · Reviewer_FQ5L · 2026-03-07

**Soundness:** 3
**Presentation:** 3
**Significance:** 2
**Originality:** 3
**Overall Recommendation:** 4
**Confidence:** 4

**Summary:**

This paper introduces ZO Fine-Tuner, a learning-to-learn zeroth-order optimizer for LLM fine-tuning that improves upon standard methods like MeZO by utilizing tiny auxiliary networks (PertNNs) to learn adaptive, block-specific perturbation variances instead of relying on fixed distributions. After a brief, one-time first-order meta-learning phase on a base model, the learned perturbation strategies successfully generalize to diverse downstream tasks

**Compliance With Llm Reviewing Policy:**

Affirmed.

**Final Justification:**

I have updated my score to 4 accordingly

**Key Questions For Authors:**

see weakness

**Limitations:**

yes

**Strengths And Weaknesses:**

Strengths：

1. The approach introduces a neat, adaptive twist to zeroth-order optimization by using tiny auxiliary networks to learn per-block perturbation variances, successfully eliminating the need for hand-crafted noise schedules.

2.  The paper effectively validates the interesting hypothesis that perturbation strategies learned during a one-time meta-training phase on a foundation model can seamlessly generalize to diverse downstream tasks and model derivatives, yielding consistent gains over baseline methods.

3. The algorithm is remarkably lightweight. Detailed runtime and memory measurements confirm that the method introduces minimal overhead compared to standard MeZO, proving its strong potential for practical, real-world deployment.

Weakness：

1.  While the paper claims to reduce reliance on hand-crafted strategies, it fails to specify whether prompt templates were utilized during training. This is a critical omission because MeZO-style methods typically suffer severe performance degradation without properly designed templates.

2. Since the contribution of the work is related to reducing the variance of the gradient estimates, the paper lacks a necessary comparison with existing variance-reduction zeroth-order methods, most notably MeZO-SVRG （without hand-crafted prompt templates ）.

3. whether you analysis how using different tasks for meta-training might impact the final performance and generalizability of the optimizer.

---

> ### Author Rebuttal · Authors · 2026-03-31
>
> We thank the reviewer for the helpful feedback. Below, we address each of the reviewer’s concerns. During the rebuttal period, we also **ran several additional experiments**, including **further ablations on the choice of meta-learning datasets** and **additional comparisons with MeZO-SVRG**. The results are also presented below.
>
> > **W1: Prompt templates**
>
> In our experiments, we used the **same prompt template as MeZO** during downstream training, following the setup described in Appendix E.2 of the original MeZO paper. Our goal was to keep the training pipeline aligned with the standard MeZO setting so that the comparison focuses on the optimizer itself rather than other artifacts. We will add this clarification to the paper.
>
> > **W2: MeZO-SVRG**
>
> We agree that MeZO-SVRG [1] is a strong baseline. In the meantime, we would like to clarify that its mechanism is largely orthogonal to ours. Our ZO-Finetuner improves the zeroth-order updates by learning a better perturbation, whereas MeZO-SVRG reduces variance by combining full-batch and mini-batch gradient estimation to stabilize optimization. In this sense, the **two methods are complementary rather than mutually exclusive**; the SVRG technique should be equally applicable on top of our methods.
>
> We have added **additional comparisons** among **MeZO**, **MeZO-SVRG**, **Our original ZO-Finetuner**, and **ZO-Finetuner-SVRG** on COPA and SST-2 for Llama-3.2-1b. The results can be found in Table C. We can see that SVRG indeed improves initial convergence speed, and **our ZO-Finetuner combined with SVRG yields superior performance to MeZO-SVRG**.
>
> **Table C: Loss comparison of MeZO, MeZO-SVRG, ZO Fine-tuner, and ZO Fine-tuner-SVRG on COPA and SST-2 with LLaMA-3.2-1B; the lower the better.**
> | Dataset | Method | 0-4000| 4000-8000| 8000-12000 | 12000-16000 | 16000-20000 |
> | --- | --- | ---: | ---: | ---: | ---: | ---: |
> | SST-2 | ZO-Finetuner (Ours) | 0.4239 | 0.2609 | **0.2231** | **0.2152** | **0.2099** |
> | | MeZO | 0.5242 | 0.4256 | 0.3835 | 0.3445 | 0.3095 |
> | | ZO-Finetuner-SVRG (Ours) | **0.3225** | **0.2357** | 0.2498 | 0.2675 | 0.3024 |
> | | MeZO-SVRG | 0.4436 | 0.3447 | 0.3111 | 0.3105 | 0.3168 |
> | COPA | ZO-Finetuner (Ours) | 2.1694 | 1.8668 | 1.7658 | 1.7175 | **1.6840** |
> | | MeZO | 2.3143 | 2.0741 | 1.9392 | 1.8490 | 1.7948 |
> | | ZO-Finetuner-SVRG (Ours) | **1.9355** | **1.7433** | **1.7212** | **1.7138** | 1.7267 |
> | | MeZO-SVRG | 2.1304 | 1.8665 | 1.7816 | 1.7484 | 1.7437 |
>
> > **W3: Meta-learning dataset**
>
> In our experiments, we chose COPA as the meta-training dataset primarily because of the smooth loss change and its small size, which makes the meta-learning process faster. This choice is made for efficiency rather than because COPA is a special or necessary design component of our method. To clarify this point, in the rebuttal, we added **additional experiments**. Specifically, we trained additional ZO Fine-tuners for LLaMA-3.2-1B using SQuAD as the meta-training dataset and for LLaMA-3.1-8B using SST-2 as the meta-training dataset. We then applied these learned optimizers to fine-tune the corresponding models on SST-2, COPA, and SQuAD. The results can be found in Tables A and B. It can be observed that the trained fine-tuner remains consistently better than MeZO. We will add this important clarification to our paper.
>
> **Table A: Loss comparison of downstream fine-tuning on SST-2, COPA, and SQuAD between MeZO and SQuAD-trained ZO Fine-tuner on LLaMA-3.2-1B; the lower the better.**
>
> | Dataset | Method | 0-4000 | 4000-8000| 8000-12000 | 12000-16000| 16000-20000 |
> | --- | --- | ---: | ---: | ---: | ---: | ---: |
> | SST-2 | SQuAD-trained finetuner | 0.4459 | 0.3143 | 0.2862 | 0.2784 | 0.2713 |
> |  | MeZO | 0.5242 | 0.4256 | 0.3835 | 0.3444 | 0.3095 |
> | COPA | SQuAD-trained finetuner | 2.1608 | 1.8948 | 1.8243 | 1.7898 | 1.7604 |
> |  | MeZO | 2.3142 | 2.0740 | 1.9392 | 1.8490 | 1.7948 |
> | SQuAD | SQuAD-trained finetuner | 0.7280 | 0.5314 | 0.5120 | 0.5017 | 0.4903|
> |  | MeZO | 0.8808 | 0.6078 | 0.5330 | 0.4957 | 0.4827 |
>
> **Table B: Loss comparison of downstream fine-tuning on SST-2, COPA, and SQuAD between MeZO and SST-2-trained ZO Fine-tuner on LLaMA-3.2-8B; the lower the better.**
>
> | Dataset | Method | 0-4000| 4000-8000| 8000-12000 | 12000-16000 | 16000-20000 |
> | --- | --- | ---: | ---: | ---: | ---: | ---: |
> | SST-2 | SST-2-trained finetuner | 0.4299 | 0.2751 | 0.2234 | 0.2003 | 0.1862 |
> |  | MeZO | 0.5166 | 0.4163 | 0.3752 | 0.3407 | 0.3060 |
> | COPA | SST-2-trained finetuner| 1.8983 | 1.5740 | 1.4699 | 1.4096 | 1.3669 |
> | | MeZO | 2.1358 | 1.9317 | 1.7904 | 1.6788 | 1.5877 |
> | SQuAD | SST-2-trained finetuner | 0.5015 | 0.3522 | 0.3275 | 0.3159 | 0.3076 |
> |  | MeZO | 0.6381 | 0.4108 | 0.3569 | 0.3384 | 0.3242 |
>
> [1] Gautam, Tanmay, et al. "Variance-reduced zeroth-order methods for fine-tuning language models." arXiv:2404.08080

---

> > ### Author Rebuttal · Reviewer_FQ5L · 2026-04-02
> >
> > N/A

---

> > > ### Author Response · Authors · 2026-04-07
> > >
> > > We are glad that our rebuttal helped address the concerns. We greatly appreciate the reviewer’s time and constructive feedback on our work!

---

### Official Review · Reviewer_jxCQ · 2026-03-07

**Soundness:** 4
**Presentation:** 4
**Significance:** 3
**Originality:** 3
**Overall Recommendation:** 5
**Confidence:** 4

**Summary:**

The paper introduces ZO Fine-tuner, a learning-to-learn zeroth-order optimizer for fine-tuning LLMs that learns adaptive, block-wise perturbation variances using lightweight per-block neural networks (PertNNs), enabling memory-efficient training comparable to inference while outperforming hand-crafted zeroth-order baselines like MeZO. It leverages first-order fine-tuning trajectories to train the optimizer once per base model, demonstrating strong generalization across 4 LLMs (LLaMA-3.2-1B, LLaMA-3.1-8B, Qwen2.5-14B, OPT-30B) and 7 datasets, achieving lower final loss in ~80% of cases and ~2.5% average accuracy gains with negligible overhead (<2MB storage for OPT-30B PertNNs).

**Compliance With Llm Reviewing Policy:**

Affirmed.

**Key Questions For Authors:**

1. How sensitive are L2L training results to trajectory generator (SGD vs. Adam; Tables 11-12 show minor differences) or dataset (COPA; Appendix D.5 on multi-dataset)? Stronger multi-task ablations could elevate significance.
2. Appendix B.1 mentions non-diagonal/low-rank extensions (e.g., with LOZO); preliminary results? Could show if block-diagonal is optimal vs. richer structures without overhead. Positive data might strengthen soundness.
3. Theorem 3.1 assumes block-diagonal Hessian (Appendix E.4); empirical Hessian/Fisher block interactions (e.g., off-diagonal norms) on your models, building on? Could solidify theory.
4. Hyperparameter grids (Table 5) small; does ZO Fine-tuner reduce tuning needs vs. baselines? Full sweeps/AutoML comparison?
5. Limitations note LoRA/quantization orthogonality; combined benchmarks (e.g., QLoRA + ZO Fine-tuner)?
6. Beyond classification/short tasks, how does it generalize to long-context?

**Limitations:**

No.
- Appendix B.1/B discusses technical limits (e.g., diagonal-only, no LoRA) but omits societal impacts (e.g., dual-use risks for easier fine-tuning)

**Strengths And Weaknesses:**

- Soundness: Strong empirical support via comprehensive experiments (e.g., Table 1 shows good outperformance; Figure 2 loss curves demonstrate faster convergence), ablation studies validating components like normalization (Table 2) and block-wise sharing (Table 3, motivated by Theorem 3.1 on block-diagonal Hessians), and theory (Appendix E) providing bounds under reasonable assumptions (e.g., local effective rank). Methods are appropriate for memory-constrained ZO fine-tuning, with honest reporting of gaps to first-order Adam (Appendix D.7, Tables 9-10).
    - Minor concern: L2L training approximates gradient flow through finite-difference (no backprop at deployment), plus unaddressed directional bias from anisotropic perturbations.

- Presentation: Well-structured with clear narrative (e.g., Figure 1 illustrates architecture; Algorithms 1/2 precise), good reproducibility details in Table 5, and positioning against baselines.
    - Minor clarity issues such as using “informal” Theorem 3.1 but defers full proof to Appendix E; hyperparameters not exhaustively swept across settings.

- Significance: Addresses key practical problem of memory-efficient LLM fine-tuning (e.g., ZO uses ~1/5th Adam memory, Table 7), with "train-once, reuse widely" paradigm unlocking utility for base model distributors; broad impact potential as it scales L2L to LLMs via block-wise design, influencing future ZO/PEFT hybrids (noted in limitations). Scope matches modest but useful gains over SOTA ZO.

- Originality: Novel L2L application to ZO-LLM fine-tuning via compact block-wise PertNNs (inspired by Transformer Hessian structure), distinguishing from prior hand-crafted ZO (e.g., HIZOO) or small-scale L2L ZO; strong justification via generalization to derivatives (Table 4). Builds creatively on MeZO but adds adaptive perturbations absent in baselines.

---

> ### Author Rebuttal · Authors · 2026-03-31
>
> We thank the reviewer for the constructive comments and the acknowledgement of our work. As part of the rebuttal, we **performed a number of new experiments**, including **additional studies on the role of the meta-learning dataset** and tests of **whether our ZO Fine-tuner transfers to a longer-sequence math dataset**. The corresponding results are summarized below.
>
> > **Q1: Trajectory Generator and Meta-Learning Dataset**
>
> In our experiments, we chose COPA as the meta-training dataset primarily because of the smooth loss change and its small size, which makes the meta-learning process faster. This choice is made for efficiency rather than because COPA is a special or necessary design component of our method. To clarify this point, in the rebuttal, we added **additional experiments**. Specifically, we trained additional ZO Fine-tuners for LLaMA-3.2-1B using SQuAD as the meta-training dataset and for LLaMA-3.1-8B using SST-2 as the meta-training dataset. We then applied these learned optimizers to fine-tune the corresponding models on SST-2, COPA, and SQuAD. The results can be found in **Tables A and B for reviewer a9nm**. It can be observed that the trained fine-tuner **remains consistently better than MeZO**. We will add this important clarification to our paper.
>
> > **Q2 & Q5: Low-Rank preliminary results and Combine with QLoRA**
>
> We agree that exploring low-rank or richer structures in the perturbation space and combining our approach with PEFT and quantization will further strengthen our paper. However, admittedly, such extensions are non-trivial and can hardly be done during the limited time during rebuttal. We will thus leave these two points as important future works to explore.
>
> > **Q3: Block-diagonal Hessian**
>
> We note that computing Hessians for LLMs is not an easy task, even for relatively small ones. Therefore, we **provide references for several papers that do observe this phenomenon** [1,2,3] for the standard transformer architectures. The models we study follow the same standard Transformer architecture, and our block partition (embeddings, Q/K/V projections, output projections, MLPs) is aligned with these architectural components.
>
> > **Q4: Hyperparameter**
>
> First, for hyperparameter tuning difficulty, we notice that, except for MeZO, all baselines require extra method-specific hyperparameters. However, after the one-time meta-learning of our ZO-Finetuner, the only hyperparameter for our method is the learning rate. Moreover, we often observe that our method achieves performance comparable to the baselines with a smaller learning rate, as illustrated in Section 4.2 of the paper. This further reduces the difficulty in hyperparameter tuning. As for more extensive hyperparameter sweeps/AutoML comparisons, we kindly note that it will require significantly more computational resources. Since our evaluation is extensive across 4 models and 7 datasets, this is unfortunately beyond our current capability.
>
> > **Q6: Generalization to long-context**
> To assess whether our method generalizes to longer sequences, we **added a new experiment on math fine-tuning**, which has a long solution and is closer to modern reasoning training. Concretely, we fine-tuned Qwen2.5-14B on MetaMathQA using the ZO Fine-tuner trained on COPA for 10000 steps, and compared it with MeZO under the same setup. The results can be found in Tables C and D, where we report the average loss during training and the final evaluation results on GSM8K and Math-500. We observe that the COPA-trained ZO Fine-tuner **still outperforms MeZO on this math dataset**. This result suggests that the learned optimizer is not restricted to the GLUE/SuperGLUE-style tasks used in the main paper, and can transfer nontrivially to a substantially different task suite.
>
> **Table C: Loss of MeZO/ZO-Finetuner during training Qwen2.5-14b on MetaMathQA; the lower the better.**
> | Training steps | MeZO | ZO-Finetuner |
> | - | -: | -: |
> | 0-2000 | 0.1978 | 0.1797 |
> | 2000-4000 | 0.1784 | 0.1604 |
> | 4000-6000 | 0.1702 | 0.1577 |
> | 6000-8000 | 0.1658 | 0.1568 |
> | 8000-10000 | 0.1646 | 0.1559 |
>
> **Table D: Evaluation results; the higher the better.**
> | Dataset | Qwen2.5-14B base | Qwen2.5-14B w/ MeZO | Qwen2.5-14B w/ Ours |
> | - | -: | -: | -: |
> | GSM8K | 78.9 | 81.4 | **85.6** |
> | Math-500 | 43.2 | 53.0 | **54.6** |
>
> > **Limitations: Societal Impacts**
>
> We will revise the Impact Statement to explicitly acknowledge such dual-use risks, including the possibility that more accessible fine-tuning may lower barriers to deploying harmful or deceptive domain-specialized models.
>
> [1] Zhang, Yushun, et al. "Adam-mini: Use fewer learning rates to gain more." arXiv:2406.16793.\
> [2] Zhang, Yushun, et al. "Why transformers need adam: A hessian perspective." NeurIPS 2024\
> [3] Ormaniec, Weronika, Felix Dangel, and Sidak Pal Singh. "What does it mean to be a transformer? insights from a theoretical Hessian analysis." arXiv:2410.10986.

---

> > ### Author Rebuttal · Reviewer_jxCQ · 2026-04-02
> >
> > The rebuttal effectively addresses the points on meta‑training dataset choice and long‑context transfer, and the additional experiments do help strengthen the empirical case. It also provides a reasonable justification for not exploring low‑rank and PEFT‑style combinations within the rebuttal window, which is understandable given the constraints. Nonetheless, the block‑diagonal Hessian assumption remains the main unresolved issue; I would encourage the authors to explicitly acknowledge this assumption’s limitations and corresponding future work in the paper, if accepted.

---

> > > ### Author Response · Authors · 2026-04-07
> > >
> > > We are glad that the rebuttal effectively addressed many of your concerns. We also agree that more empirical validations on the block-diagonal Hessian structure will further strengthen our work. If accepted, we will make this point more explicit in the paper and highlight it as an important direction for future work. Thank you again for the constructive comments.

---

### Official Review · Reviewer_a9nm · 2026-03-11

**Soundness:** 3
**Presentation:** 3
**Significance:** 2
**Originality:** 3
**Overall Recommendation:** 4
**Confidence:** 4

**Summary:**

This work extends MeZO, a popular zero-th order optimizer for finetuning LLMs. MeZO performs $\mathcal{N}(0, \mathbf{I})$ perturbations, while ZO fine-tuner proposes to use learned covariances of the perturbations via meta-learning. To efficiently parameterize the meta-network, they borrow ideas from Adam-mini to localize the covariances based on the observation from previous works that the Hessian follows a block-diagonal structure. The paper studies the proposed optimizer on a suite of classification-style natural language understanding (NLU) tasks.

**Compliance With Llm Reviewing Policy:**

Affirmed.

**Final Justification:**

The additional results presented in the rebuttal are strong in the fine-tuning settings.

I have updated my score to 4 accordingly. I do not have any more questions to the authors, but refrain from going even higher on my score mainly because of limited evaluation on the currently highly relevant area of post-training style fine-tuning which I believe would have strenghthened this work even more.

**Key Questions For Authors:**

See Weaknesses

**Limitations:**

N/A

> We believe these methods can be combined with ZO Fine-tuner to improve the memory-performance tradeoff in practice further.

These claims are not necessarily trivial, and may be a misleading claim to make without empirical proof (which is left for future work). Techniques such as PEFT, Quantization may induce additional training instabilities which learned optimizers may be more sensitive to, and this needs to be acknowledged.

**Strengths And Weaknesses:**

Soundness:
1. The idea is simple and easy to understand. It is very clear why meta-learning of specific covariances of perturbations can be better than a simple identity covariance.
2. “ZO Fine-tuner trained on a single dataset is highly generalizable across model derivatives and datasets”. This is a very strong claim to make with just a single dataset (COPA) used to validate this on a set of related natural language understanding tasks that come from the same suite (GLUE/SuperGLUE). This claim is only valid if training the meta optimizer on each of the individual tasks can help fine-tune on the other tasks.
3. Assumptions needed to be made about the meta-learning dataset. “This choice is mainly due to COPA’s consistently smooth loss decrease during standard fine-tuning”. I do not fully see why this is a reasonable assumption to make and intuition for how a model provider can choose the meta-learning dataset effectively.


Presentation:
1. The approach, training algorithms, and results are presented well. Some key hyper parameters of the optimizer are ablated.
2. My main concerns are with the limited presentation of the style of tasks studied that falls short of the wide range of fine-tuning paradigms followed in 2025/2026.

Significance:
1. Nowadays, more relevant tasks for LLMs are longer, harder generation tasks such as instruction tuning, reasoning, code generation as these are the frontier of current fine-tuning regimes. It's unclear whether this approach is going to do well there. The strength of MeZO and other FO optimizers is that they require no assumption about the downstream fine-tuning task. For example, it's unclear if learning a ZO optimizer on COPA will do well on Math finetuning. Downstream practitioners who may fine tune on very diverse domains (that go beyond the closed suite of datasets used in this work that follow similar task structure) may prefer certain guarantees about the optimizer. MeZO or FO optimizers require no such assumptions.
2. For example, RL is currently a dominant paradigm. MeZO, or any other FO optimizer in principle can be applied to RL or SFT or any other post training paradigm without any extra modification. This method requires learning an optimizer for RL (or any other post-training paradigm not seen by the optimizer) separately and could limit widespread adoption.


Novelty:
1. Combines insights from Adam-mini ( with that of MeZO). Formulating this combination as a meta-learning problem is novel to the best of my knowledge.

---

> ### Author Rebuttal · Authors · 2026-03-31
>
> We thank the reviewer for the thoughtful comments. Below, we address each of the reviewer’s concerns. During rebuttal, we also **conducted several new experiments**, including **more ablations on the meta-learning dataset** and **generalizability of our ZO-Finetuner to a longer sequence math dataset**, whose results are summarized below.
>
> > **Soundness 3: Meta-Learning Dataset**
>
> In our experiments, we chose COPA as the meta-training dataset primarily because of the smooth loss change and its small size, which makes the meta-learning process faster. This choice is made for efficiency rather than because COPA is a special or necessary design component of our method. To clarify this point, in the rebuttal, we added **additional experiments**. Specifically, we trained additional ZO Fine-tuners for LLaMA-3.2-1B using SQuAD as the meta-training dataset and for LLaMA-3.1-8B using SST-2 as the meta-training dataset. We then applied these learned optimizers to fine-tune the corresponding models on SST-2, COPA, and SQuAD. The results can be found in Tables A and B. It can be observed that the trained fine-tuner remains consistently better than MeZO. We will add this important clarification and additional results to our paper.
>
> **Table A: Loss comparison of downstream fine-tuning on SST-2, COPA, and SQuAD between MeZO and SQuAD-trained ZO Fine-tuner on LLaMA-3.2-1B; the lower the better.**
>
> | Dataset | Method | 0-4000 | 4000-8000| 8000-12000 | 12000-16000| 16000-20000 |
> | --- | --- | ---: | ---: | ---: | ---: | ---: |
> | SST-2 | SQuAD-trained finetuner | 0.4459 | 0.3143 | 0.2862 | 0.2784 | 0.2713 |
> |  | MeZO | 0.5242 | 0.4256 | 0.3835 | 0.3444 | 0.3095 |
> | COPA | SQuAD-trained finetuner | 2.1608 | 1.8948 | 1.8243 | 1.7898 | 1.7604 |
> |  | MeZO | 2.3142 | 2.0740 | 1.9392 | 1.8490 | 1.7948 |
> | SQuAD | SQuAD-trained finetuner | 0.7280 | 0.5314 | 0.5120 | 0.5017 | 0.4903|
> |  | MeZO | 0.8808 | 0.6078 | 0.5330 | 0.4957 | 0.4827 |
>
> **Table B: Loss comparison of downstream fine-tuning on SST-2, COPA, and SQuAD between MeZO and SST-2-trained ZO Fine-tuner on LLaMA-3.2-8B; the lower the better.**
>
> | Dataset | Method | 0-4000| 4000-8000| 8000-12000 | 12000-16000 | 16000-20000 |
> | --- | --- | ---: | ---: | ---: | ---: | ---: |
> | SST-2 | SST-2-trained finetuner | 0.4299 | 0.2751 | 0.2234 | 0.2003 | 0.1862 |
> |  | MeZO | 0.5166 | 0.4163 | 0.3752 | 0.3407 | 0.3060 |
> | COPA | SST-2-trained finetuner| 1.8983 | 1.5740 | 1.4699 | 1.4096 | 1.3669 |
> | | MeZO | 2.1358 | 1.9317 | 1.7904 | 1.6788 | 1.5877 |
> | SQuAD | SST-2-trained finetuner | 0.5015 | 0.3522 | 0.3275 | 0.3159 | 0.3076 |
> |  | MeZO | 0.6381 | 0.4108 | 0.3569 | 0.3384 | 0.3242 |
>
> > **Presentation 2 & Soundness 2 & Significance 1: Generalization to the math domain**
>
> To assess whether our method generalizes to longer-sequence and more modern SFT settings, we **added a new experiment on math fine-tuning**. Concretely, we fine-tuned Qwen2.5-14B on **MetaMathQA** using the ZO Fine-tuner trained on COPA for 10000 steps, and compared it with MeZO under the same setup. The results can be found in Tables C and D, where we report the average loss during training and the final evaluation results on GSM8K and Math-500. We observe that the COPA-trained ZO Fine-tuner **still significantly outperforms MeZO** on this math dataset. This result suggests that the learned optimizer is not restricted to the GLUE/SuperGLUE-style tasks used in the main paper, and can transfer nontrivially to a substantially different task suite.
>
> **Table C: Loss of MeZO/ZO-Finetuner during training Qwen2.5-14b on MetaMathQA; the lower the better.**
> | Training steps | MeZO | ZO-Finetuner |
> | --- | ---: | ---: |
> | 0-2000 | 0.1978 | 0.1797 |
> | 2000-4000 | 0.1784 | 0.1604 |
> | 4000-6000 | 0.1702 | 0.1577 |
> | 6000-8000 | 0.1658 | 0.1568 |
> | 8000-10000 | 0.1646 | 0.1559 |
>
> **Table D: Evaluation results; the higher the better.**
> | Dataset | Qwen2.5-14B base | Qwen2.5-14B w/ MeZO | Qwen2.5-14B w/ Ours |
> | --- | ---: | ---: | ---: |
> | GSM8K | 78.9 | 81.4 | **85.6** |
> | Math-500 | 43.2 | 53.0 | **54.6** |
>
> > **Significance 2: applicability to RL**
>
> We agree that zeroth-order methods could in principle be applied to RL. However, for LLM post-training, this direction is still largely open, especially for modern reasoning-RL methods such as GRPO. We therefore regard this as a promising future direction rather than something the current paper aims to solve.
>
> > **Limitations**
>
> Thank you for pointing this out. We will explicitly add acknowledgements about the difficulty of further extensions regarding integrating our methods with PEFT techniques or quantization approaches for further memory savings.

---

> > ### Author Rebuttal · Reviewer_a9nm · 2026-04-01
> >
> > I thank the authors for addressing most of my concerns. The results presented in the rebuttal are strong in the fine-tuning settings.
> >
> > I have updated my score to 4 accordingly. I do not have any more questions to the authors, but refrain from going even higher on my score mainly because of limited evaluation on the currently highly relevant area of post-training style fine-tuning which I believe would have strenghthened this work even more.

---

> > > ### Author Response · Authors · 2026-04-07
> > >
> > > We are glad that the additional results helped address the main concerns, and we agree that broader evaluation in this highly relevant setting would further strengthen the work. We will highlight this more clearly as an important direction for future study. Thank you again for your constructive feedback!

---

### Official Review · Reviewer_x315 · 2026-03-12

**Soundness:** 2
**Presentation:** 3
**Significance:** 3
**Originality:** 3
**Overall Recommendation:** 4
**Confidence:** 4

**Summary:**

The paper proposes ZO Fine-tuner, a zeroth-order optimizer for LLM finetuning whose perturbation distribution is proposed by a small neural network learned over real data. Here the sampling distribution takes the parameteric form of zero mean gaussian with a block diagonal covariance matrix where each block is an individually scaled identity matrix. For each parameter block, the small neural network takes in the  summary statistics of the parameters’ mean and variance in addition to the last iteration’s loss and previously proposed sampling standard deviation and outputs the new sampling standard deviation. The authors then propose a one-time per-model training algorithm for ZO Fine-tuner where the neural network is updated by the reparametrization gradient of the one-step updated model’s loss with respect the network parameters. Empirically, the authors demonstrate the ZO Fine-tuner outperforms prior human designed ZO baselines and can generalize to different fine-tuning tasks despite being trained on others.

**Compliance With Llm Reviewing Policy:**

Affirmed.

**Key Questions For Authors:**

- How is $\eta_1$ chosen in Algorithm 2? I think $\eta_1$ can potentially affect the learned neural network’s output’s magnitude.
- is there a reason for using SGD to update $\omega_t$ and $\theta_t$? I would think one could use better optimizers to optimize $\omega_t$. On the other hand, to make the optimizer more on-policy, maybe the LLM parameter could also be actually optimized by the FineTuner’s sampled gradient estimate a few lines above? I wonder if the authors have thought more about this.
- Does the cross dataset generalization work for other dataset other than COPA?
- How does the trajectory of PertNN’s output sigmas change? Also how does the inputs to the neural network change throughout training?

**Limitations:**

Yes.

**Strengths And Weaknesses:**

## Soundness
I think the proposed method is generally sound. Two things I hope the authors can give better explanations are:

1. ZO Fine-tuner’s gradient estimate is not unbiased with respect to the smoothed loss. The MEZO gradient estimator can be derived as an unbiased gradient with respect to the smoothened loss under isotropic gaussian smoothing (through likelihood ratio gradient). By a similar analogy, I would by default assume the authors’ proposed gradient estimator is the likelihood ratio gradient under the gaussian smoothing with block diagonal anistropic covariance. However, the proposed estimator does not have the proper $1/\sigma_i$ scaling for each corresponding block. Thus I believe the estimator is not unbiased. I think this makes the theory analysis more difficult, and the authors as a compromise turns to analyze a block-by-block update method to maintain the unbiased. I wonder if the authors have considered the unbiased estimator formulation and if there are specific reasons to not use it.

2. Why do we believe the inputs to the PertNN are sufficient? It seems surprising to me that the mean and variance of the parameter block along with only the current iterations’ losses and the previously proposed standard deviation are sufficient for the neural network to propose good sampling standard deviation. Can the authors provide any explanations of why this is reasonable except for computational reasons. (I would naively think the neural network needs to see at least a sequence of realized losses and proposed standard deviations to have somes notion of under what sampling distributions can the model make progress.


## Presentation:
The paper’s presentation has good presentation and generally reads well.

## Significance
I think the experimental results of ZO Fine-tuner outperforming other Zeroth-order optimizer is an interesting result. I also find the fact that ZO Fine-tuner trained just on the COPA dataset can generalize to other fine-tuning tasks surprising and potentially significant. One thing I hope the authors can further clarify is the computational cost of training the ZO optimizer. I understand the authors have talked about the cost of running the learned ZO optimizer (Algorithm 1), but I’m not sure if the paper has discussed how much compute it takes to train a ZO optimizer (Algorithm 2). The training requires computing multiple LLM loss evaluations as well as backprop to compute the full parameter gradient. I think describing this cost makes it more obvious how much extra cost the model provider needs to take on to provide this additional artifact.

## Originality
Even though the idea of learning an optimizer through data is no longer novel, applying it to learn a zeroth-order fine-tuning optimizer is an interesting and (to the best of my knowledge) under-explored direction.

---

> ### Author Rebuttal · Authors · 2026-03-31
>
> We thank the reviewer for their feedback. Below, we address each of the reviewer’s concerns. In addition, we conducted several new experiments during the rebuttal, including **more ablations on the meta-training dataset**, and the corresponding results are also reported below.
>
> > **S1: Unbiased Gradient**
>
> We agree with the reviewer that the proposed method does not, in general, yield an unbiased gradient estimator, and that unbiasedness is appealing from a theoretical perspective. However, in zeroth-order optimization, practical performance depends not only on bias but also on variance, and our design should be viewed through this bias-variance tradeoff. In particular, if the learned block-wise $\Sigma$ allocates more perturbation mass to useful directions and less to less informative ones, then the resulting estimator can have substantially lower variance and yield more stable and effective updates in practice, even if it is biased.
>
> > **S2: Inputs**
>
> We would like to clarify that our current design captures a one-step history feedback: PertNN observes the previous variance choice together with the resulting losses, so it can adapt the next block-wise variance allocation based on the immediate effect of the last perturbation. For instance, if the previous perturbation allocation leads to less favorable loss behavior, PertNN can modify the next variance pattern, which in turn changes the distribution of sampled perturbation directions. While this is less expressive than a design with longer memory, **we believe our strong empirical results suggest that this compact one-step feedback mechanism is already effective in practice**.
>
> > **Significance: Cost of L2L**
>
> We have analyzed the computation cost of L2L in Appendix D.4 of the paper. In our experiments, L2L takes approximately 2.5x memory cost and 2.4x time cost of a standard first-order fine-tuning pass. We believe this computation overhead is still acceptable as it **only needs to be done once**.
>
> > **Q1: Choice of eta**
>
> A detailed hyperparameter choice in our L2L framework is introduced in the paper Appendix C.3. Empirically, for our experiments, we chose $\eta_1 = 1e^{-6}$, which ensures smooth loss decrease when fine-tuning with SGD.
>
> > **Q2: Choice of using SGD**
>
> We have ablation results comparing the trained ZO-Finetuner with SGD and Adam. The results can be found in the paper’s Appendix Tables 11 and 12. We observed that the downstream fine-tuning performance remains similar. Therefore, we chose to use SGD since the corresponding memory footprint is lower compared to better optimizers.
>
> We have also considered directly updating the LLM parameters using the ZO Fine-tuner’s sampled gradient estimate in order to make the procedure more on-policy. However, in our preliminary experiments, this led to unstable training. Intuitively, at the beginning of training, the learned optimizer is still weak, so the sampled zeroth-order updates may not reliably reduce the LLM loss. As a result, the model trajectory can become stagnant, which in turn makes it harder for the fine-tuner itself to improve. In contrast, a first-order optimizer provides a much smoother and more reliable loss descent trajectory, which exposes the fine-tuner to model states spanning a broad range of loss values and supplies a more stable learning signal during meta-training. We found this diversity of training states to be important for successfully training the fine-tuner.
>
> > **Q3: Meta-Learning Dataset**
>
> We **conducted additional experiments** by training additional ZO Fine-tuners for LLaMA-3.2-1B using SQuAD as the meta-training dataset and for LLaMA-3.1-8B using SST-2 as the meta-training dataset. We then applied these learned optimizers to fine-tune the corresponding models on SST-2, COPA, and SQuAD. The results can be found in **Tables A and B for reviewer a9nm**. It can be observed that the trained fine-tuner **remains consistently better than MeZO**, which indicates that the cross-dataset generalization did not just magically happen on COPA, but **can be achieved on all datasets**.
>
> > **Q4: Trajectory and Input of PertNN**
>
> We thank the reviewer for this question. At present, we do not yet have a fully developed intuition for how PertNN’s outputs evolve during downstream use, and we mainly treat the learned ZO Fine-tuner as a black-box optimizer. That being said, our experiments do confirm that both the per-block variances and the relative variance ratios across blocks change over optimization steps. As a result, both the overall magnitude and the effective direction of the sampled perturbation change from step to step. We will include some informative plots and related discussions in our revised paper. A more detailed analysis of these dynamics would be very interesting, and we view it as an important future work.

---

> > ### Author Rebuttal · Reviewer_x315 · 2026-04-03
> >
> > I would like to thank the authors for their response, which have answered some of my questions. Some of my questions that were not fully addressed are:
> > 1. Regarding unbiasedness, the authors argue that sometimes unbiasedness is not necessary, which I agree. However, I would expect the authors to experiment with the unbiased gradient estimator under the block diagonal perturbation sampling distribution. Besides, the paper has theoretically analyzed an unbiased gradient estimator (which updates the parameter blocks one block at a time) which is different from the biased estimator the authors have proposed.
> > 2. Regarding the input to PerturbNN, I don't think the authors have explained how the mean and variance of the parameter block could be useful for the sampling distribution's variance.
> > 3. Q2 The authors have explained why they use SGD to update the language model parameters in Algorithm 2, but I couldn't find the reasoning for why PertNN is updated with SGD instead of a more modern optimizer.
> >
> > I generally remain positive about the paper and will keep my current score.

---

> > > ### Author Response · Authors · 2026-04-07
> > >
> > > We thank the reviewer for further feedback and the positivity towards our work. We further provide explanations to the reviewer’s questions as follows.
> > >
> > > > **Q1: Unbiasedness**
> > >
> > > Our current theoretical analysis studies the unbiased block-wise formulation because it is more difficult to analyze the actual biased estimator we are currently using. We agree that a more systematic investigation of the biased versus unbiased formulations, both theoretically and empirically, is an interesting and important direction for future work, and we will clarify this distinction more explicitly in the revision.
> > >
> > > > **Q2: Input to PertNN**
> > >
> > > The overall magnitude and spread of the parameters in a parameter block could play a role in determining the optimal perturbation. Intuitively, if a parameter block has an overall small scale and a small spread (small mean/variance), then that block may be less able to tolerate large perturbations and vice versa.
> > >
> > > > **Q3: SGD vs. Adam on updating PertNN**
> > >
> > > We thank the reviewer for pointing this out. We conducted additional experiments comparing Adam and SGD for updating PertNN on LLaMA-3.1-8B, and report the downstream performance below in Table C. We observe that the final converged performance is mostly similar. We therefore choose SGD also for updating PertNN, due to its lower memory overhead.
> > >
> > > **Table C: Downstream performance of ZO-Finetuner trained with SGD/Adam on Llama-3.1-8B**
> > > | Dataset | Method | 0-4000 | 4000-8000 | 8000-12000 | 12000-16000 | 16000-20000 |
> > > | --- | --- | ---: | ---: | ---: | ---: | ---: |
> > > | COPA | SGD | 1.9207 | 1.5722 | 1.4589 | 1.4004 | 1.3617 |
> > > | | Adam | 2.0010 | 1.6842 | 1.5441 | 1.4460 | 1.3895 |
> > > | SST-2 | SGD | 0.4652 | 0.3146 | 0.2409 | 0.2033 | 0.1813 |
> > > | | Adam | 0.4979 | 0.3418 | 0.2602 | 0.2121 | 0.1860 |
> > > | SQuAD | SGD | 0.4921 | 0.3395 | 0.3165 | 0.3091 | 0.3065 |
> > > | | Adam | 0.5170 | 0.3447 | 0.3185 | 0.3066 | 0.3016 |

---

### Decision · Program_Chairs · 2026-04-30

**Decision:**

Accept (regular)

**Comment:**

The authors propose learning adaptive, block-specific perturbation variances via lightweight auxiliary neural networks. By executing a one-time, first-order meta-training phase on a base model, the learned optimizer can be transferred to various downstream tasks and model derivatives. The reviewer consensus is unanimously positive, with final scores leaning toward Accept/Weak Accept. The committee agrees that the paper is technically sound, well-presented, and addresses a highly relevant problem for the community: memory-efficient LLM fine-tuning.